# Cluster and Predict Latent Patches for Improved Masked Image Modeling

**Timothée Darcet**  *timdarcet@meta.com*
*Meta FAIR*
*Univ. Grenoble Alpes, Inria, CNRS, Grenoble INP, LJK*

**Federico Baldassarre**  *baldassarre@meta.com*
*Meta FAIR*

**Maxime Oquab**  *qas@meta.com*
*Meta FAIR*

**Julien Mairal**  *julien.mairal@inria.fr*
*Univ. Grenoble Alpes, Inria, CNRS, Grenoble INP, LJK*

**Piotr Bojanowski**  *bojanowski@meta.com*
*Meta FAIR*

**Reviewed on OpenReview:** *https://openreview.net/forum?id=Ycmz7qJxUQ*

## Abstract

Masked Image Modeling (MIM) offers a promising approach to self-supervised representation learning, however existing MIM models still lag behind the state-of-the-art. In this paper, we systematically analyze target representations, loss functions, and architectures, to introduce CAPI – a novel pure-MIM framework that relies on the prediction of latent clusterings. Our approach leverages a clustering-based loss, which is stable to train, and exhibits promising scaling properties. Our ViT-L backbone, CAPI, achieves 83.8% accuracy on ImageNet and 32.1% mIoU on ADE20K with simple linear probes, substantially outperforming previous MIM methods and approaching the performance of the current state-of-the-art, DINOv2. We release all our code and models.

## 1 Introduction

Recent advances in large-scale visual representation learning have established foundation models as a cornerstone of modern computer vision. Self-supervised representations have proven particularly effective in domains with limited annotations, such as satellite imagery (Tolan et al., 2024) and medical imaging (Xu et al., 2024; Vorontsov et al., 2024; Chen et al., 2024; Moutakanni et al., 2024; Dermyer et al., 2025), while enabling breakthroughs in more fundamental vision tasks like monocular depth estimation (Yang et al., 2024a; Bochkovskii et al., 2024; Yang et al., 2024b), keypoint matching (Edstedt et al., 2024), and tracking (Tumanyan et al., 2025). This shift from small supervised models to large self-supervised generalist models mirrors the evolution in natural language processing since the publication of BERT (Devlin et al., 2018) and GPT (Radford et al., 2018), where large-scale models pretrained on web-scale unlabeled data have become ubiquitous foundation models.

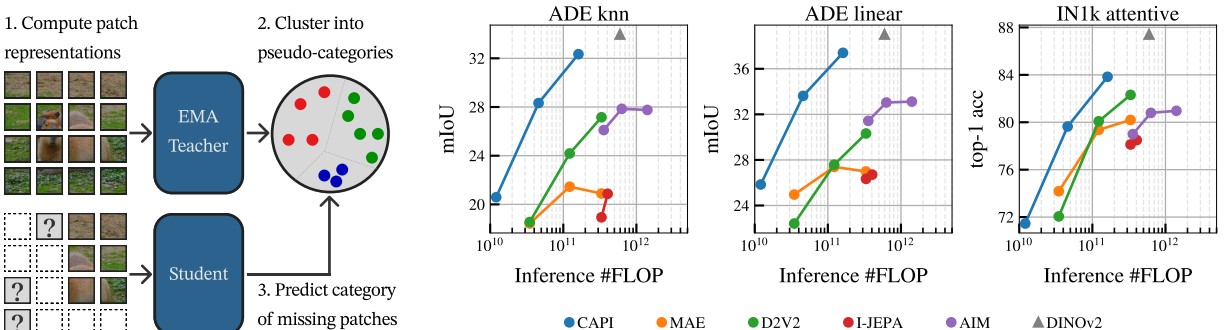

Figure 1: **CAPI Method overview:** image patches embedded by a EMA encoder are grouped into clusters. Their assignments are then used as the training signal for the online model. The loss is purely about predicting the content of missing patches and does not rely on augmentations or a contrastive loss. **Evaluation scores:** we evaluate frozen representations on ADE20K segmentation with a $k$-nn and linear probe and on ImageNet-1k classification with an attentive probe. We compare to MAE, data2vec 2.0, I-JEPA, and AIM. Compared to other masked image models, CAPI achieves higher performance with fewer FLOP, scaling well with model size, and approaches the scores of DINOv2+reg.

However, the impressive scalability of language models remains unmatched in vision: the best self-supervised visual encoders contain around one billion parameters (Oquab et al., 2024), hundreds of times smaller than current language models (Liu et al., 2024). A reason for this gap may be found in the discrepancy between the pretraining tasks used in vision and in language. The success of language models stems from the generality of the language modeling task: modeling the distribution of data, conditioned on context. Naturally, researchers have attempted to adapt this approach to computer vision, which resulted in masked image modeling (MIM): the prediction of missing image content given surrounding context.

Yet, existing MIM approaches have not matched the representation quality of alternative self-supervised methods. Pixel-level reconstruction objectives, *e.g.* MAE (He et al., 2022), provide good initialization for fine-tuning, but yield poor frozen representations, possibly because the target is too low-level to capture the task's inherent uncertainty (LeCun, 2022; Assran et al., 2023). Instead of reconstructing pixels, other works propose reconstructing in a pretrained encoder's latent space (Zhang et al., 2022; Fang et al., 2023; 2024). However, this approach requires an existing encoder and cannot learn representations from scratch.

The most promising direction uses the latent representation of the *online* model – or an exponential moving average (EMA) of it – as a learning target for the model, bootstrapping an informative latent space from scratch. This approach is used in the current state-of-the-art method for self-supervised learning of representations, DINOv2 (Oquab et al., 2024). However, these methods often suffer from poor stability (Zhou et al., 2022) and sensitivity to hyperparameters (Assran et al., 2023), requiring additional objectives like contrastive learning to produce competitive representations (Zhou et al., 2022; Oquab et al., 2024; Alkin et al., 2024).

In this work, we focus on online latent masked image modeling. We isolate it from other stabilizing objectives, and systematically study the design choices it involves. We center the discussion around three aspects of the masked image modeling principle: the target representation, the formulation of the loss, and the architecture used to perform predictions. We show that with the right implementation, a simple masked image modeling objective can lead to features competitive with the current state of the art in SSL. In brief, our method relies on a pair of vision transformers, an online model trained with masked image modeling, and a target encoder updated via an exponential moving average of the online model (fig. 1 left). The training signal for the online model comes from the patch representations of the target encoder, which is updated through an exponential moving average (EMA). The patch embeddings of the EMA encoder are converted to soft assignments on pseudo-categories using a learned online clustering, while the student receives a partially masked image as input and is trained to predict the clustering assignments of the missing patches. Using this method, we train CAPI, a 300-million parameter visual encoder whose representations approach DINOv2's performance while significantly outperforming previous masked image models (fig. 1 right).

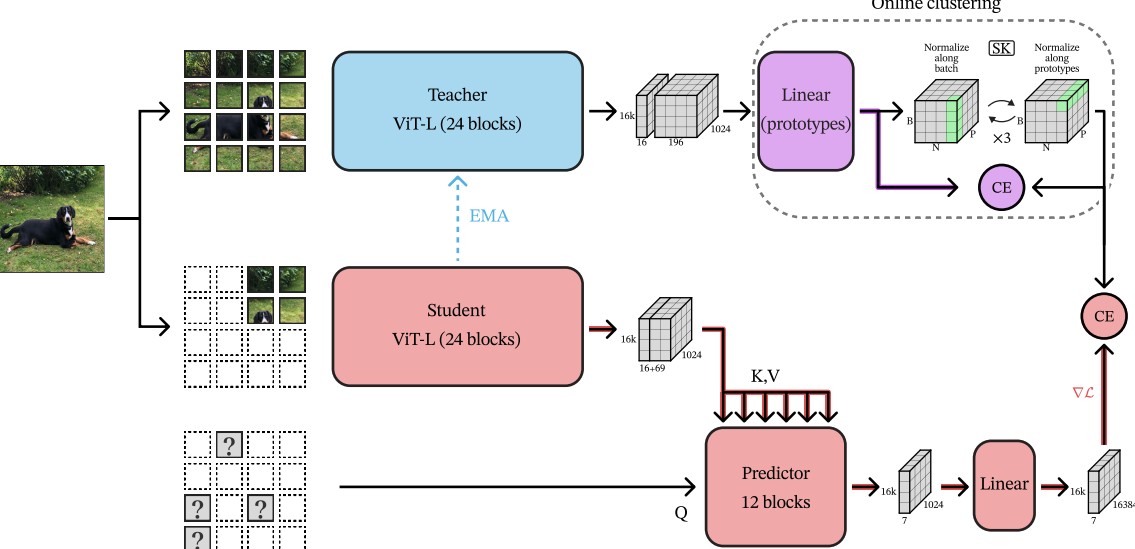

Figure 2: Detailed overview of our method with reference tensor sizes for a ViT-L/16 model. We denote in red the parts that are trained by the main loss, in purple the parts that are trained with the clustering loss, and in blue the parts that are updated by the EMA.

## 2 Related Work

**Self-supervised representation learning.** Self-supervised learning (SSL) is a pre-training paradigm in which a model is optimized to solve a pretext task on unlabeled data, often collected at scale. As the model is not trained to match specific human annotations, the benefit of this type of training is to produce a generalist model, that can be adapted to solve many different downstream tasks. Depending on the application, these tasks can be solved either by fully fine-tuning the mode, or by using the representations extracted with the frozen model. In the SSL literature, some works have focused mainly on full fine-tuning (He et al., 2022; Huang et al., 2023), while others have shown that frozen representations can reach excellent performance on a wide range of tasks, avoiding costly fine-tuning (Oquab et al., 2024) and generalizing to annotation-scarce domains where fine-tuning is not possible (Tolan et al., 2024; Xu et al., 2024). Historically, early work on self-supervised learning focused on hand-crafted pretext tasks such as predicting the rotation of an image (Gidaris et al., 2018), the relative position of patches (Doersch et al., 2015) or the color of a grayscale image (Zhang et al., 2016). Subsequent works pushed the field forward with methods based on clustering (Caron et al., 2018; Asano et al., 2020; Caron et al., 2020; 2021) and contrastive learning (Oord et al., 2018; Misra & Maaten, 2020; Chen et al., 2020b; He et al., 2020). Nowadays, the best self-supervised encoder inherits from these families (Oquab et al., 2024) and complements them with a masked image modeling objective (Zhou et al., 2022). However, training with both global and MIM objectives can prove difficult, as multiple components can interact negatively[1]. In this work, we study the masked image modeling component in isolation, suggesting improvements to properly stabilize the optimization objective in the absence of a global term.

**Pixel reconstruction.** Learning by predicting a missing part of an image was first proposed in Context encoders (Pathak et al., 2016). This was envisioned as conceptually similar to denoising autoencoders (Vincent et al., 2008), in the case where the noise is a masking process. More recently, the success of masked language modeling (Devlin et al., 2018) and autoregressive pretraining in natural language processing (Radford et al.,

---

[1]Zhou et al. (2022) showed that using a shared head for the DINO and iBOT losses gave better results at small scales. However, at large scale, Oquab et al. (2024) observed that this caused the iBOT loss to explode late into the training, significantly degrading performance. Untying the two heads prevented the two losses from competing directly, stabilizing the training.

2018) brought a new wave of interest for transferring these ideas to vision. iGPT (Chen et al., 2020a) was the first effort to train a transformer (Vaswani et al., 2017) by generating pixels. Chen et al. (2020a) proposed rasterizing images to very low resolution, then training for autoregressive next-token prediction. Then, the advent of the ViT (Dosovitskiy et al., 2021) architecture sparked further interest in the field. Following the initial exploration of Dosovitskiy et al. (2021), BeiT (Bao et al., 2021) tried using the quantized latents of a dVAE as targets for a masked image pretraining, using the tokenizer from DALL·E (Ramesh et al., 2021). BeiT has proven useful as an initialization for further fine-tuning, but severely underperformed baselines in representation learning. To simplify BeiT, SimMIM, and MAE (Xie et al., 2021; He et al., 2022) concurrently proposed using raw pixels as targets. Thanks to a clever split encoder/decoder architecture, MAE proved very stable and reached interesting representations, despite its simplicity. However, it still fell short of previous SSL methods in terms of representation quality: MAE models need to be scaled to a ViT-H size to match the linear probing performance of a $25\times$ smaller DINOv1 ViT-S/16(Caron et al., 2021).

**Latent reconstruction.** Concurrently to MAE, Zhou et al. (2022) proposed iBOT. To obtain a more semantic tokenization, iBOT used the online output of the model being trained as the reconstruction target for masked image modeling. This led to good representations, and the method was reused to obtain the current state-of-the-art (Oquab et al., 2024). However, the iBOT objective was very unstable and required an additional DINO (Caron et al., 2021) loss to stabilize the training. The idea of using online representations as targets was then reused in data2vec (Baevski et al., 2022) and I-JEPA (Assran et al., 2023; Bar et al., 2024). I-JEPA in particular proposed reusing the split encoder/decoder architecture of MAE (with the "decoder" being called "predictor" in I-JEPA) and removing the projection head of iBOT, to obtain a more stable objective in the latent space. The improvements in I-JEPA established a new tradeoff between stability and performance, but it was still both sensitive to hyperparameters ($-12$ points on IN-1k when changing the "target scale" from $[0.15, 0.2]$ to $[0.125, 0.2]$ (Assran et al., 2023)) and weaker than DINOv2 (81.1 on ImageNet-1k, 5.4 below DINOv2 while using a model twice bigger).

**Clustering in self-supervised learning.** Our approach is also related to methods that use clustering for self-supervised learning. First of this line of work, DeepCluster (Caron et al., 2018) proposed using a simple $k$-means to obtain pseudo-labels. Subsequently, SwAV (Caron et al., 2020) introduced an online clustering to replace the offline $k$-means. DINO (Caron et al., 2021) built on SWaV, and made the clustering be implicitly learned by the MLP projection head of the student. In a masked image modeling setup, iBOT (Zhou et al., 2022) proposed to reuse the DINO projection head for masked image modeling, implicitly using a clustering as target. Other recent approaches have used clustering for masked image modeling. Gidaris et al. (2024) used codebook assignments as target for a masked image modeling training, and complement it with a global loss, obtaining a formulation similar to iBOT where the clustering aspect appears more clearly. The codebook, however, is obtained by randomly sampling previous batches, whereas we attempt to learn a more optimal codebook with an online clustering. Most similar to our approach is the work of Salehi et al. (2024). They proposed a masked video modeling setup where the target encoder is a patch-wise MLP trained via gradient descent at the same time as the predicting model, via a loss comparing the target representation to the prediction of the other model. To prevent collapse, they use a Sinkhorn-Knopp algorithm to rebalance the assignments of the patches to the prototypes, obtaining a form of clustering. Both the clustering aspects and the Sinkhorn-Knopp algorithm are similar to our approach, but the target encoder is significantly different. In our study, we focus on cases where the target is the latent representation of the online model or its EMA, while they learn an entirely separate MLP to encode the targets. We also focus on images, while they focus on videos.

## 3 Approach

At a high level, masked image modeling involves masking a part of the input, feeding the visible region to a prediction model, and optimizing it to predict the content of the missing parts. Despite the simple formulation, the effectiveness of reconstruction-based methods and the properties of the trained models are dramatically influenced by a number of design choices. In this section, we discuss the three aspects of masked image modeling depicted in Figure 3: the type of patch representation used as target (fig. 4), the loss

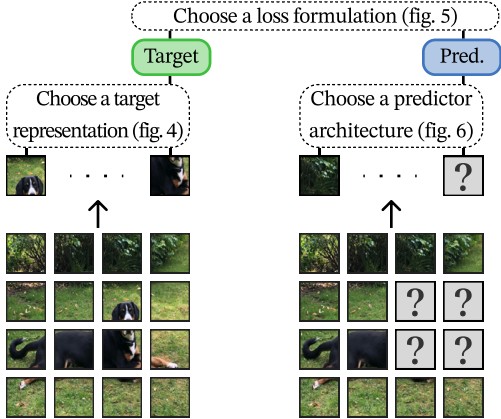

Figure 3: Overview of the components of a reconstruction-based model. We identify three main choices involved in designing a masked image model: the choice of targets (fig. 4), the loss function (Section 3.1, fig. 5) and the architecture of the predictor (Section 3.2, fig. 6).

formulation (Section 3.1, fig. 5), and the prediction architecture (Section 3.2, fig. 6). Based on our findings, we introduce CAPI, an SSL model that enjoys stable learning and strong representation capabilities.

**Overview of training.** In short, our main design choices are: first, we reconstruct images in latent space with a teacher-student framework, following iBOT. As the online latent MIM framework has already shown very promising results, we restrict our focus to this specific setup. Then, we formulate our loss using a clustering component, inspired by SeLa and SwAV, and draw inspiration from the regularization methods they introduce, in particular the Sinkhorn-Knopp (Sinkhorn & Knopp, 1967) rebalancing from SeLa (Asano et al., 2020). Finally, we employ a cross-attention predictor model, separate from the encoder, to perform reconstructions, as in crossMAE (Fu et al., 2024).

Our encoder and the predictor are transformers (Dosovitskiy et al., 2021) and, during pre-training, they operate in tandem as the student (fig. 1 left). The teacher is an EMA of the encoder. The typical values for one training iteration using square images of side 224 pixels and a model with patch size 16 are:

- We pass the full image to the teacher, collect $n = 14 \times 14 = 196$ patch tokens, and apply an online clustering to obtain soft assignments that will be used as learning targets.
- The encoder receives a partial view of the input image: we apply a patch embedding layer to obtain $n$ patch tokens, we drop $p_{\mathrm{drop}} \times n$ of these patches and pass to the encoder the remaining $n_{\mathrm{keep}} = (1 - p_{\mathrm{drop}}) \times n = 69$ patches ($p_{\mathrm{drop}} = 65\%$).
- The encoder takes these $n_{\mathrm{keep}} = 69$ tokens along with $n_{reg} = 16$ learnable register tokens (Darcet et al., 2024), and processes them to obtain $n_{\mathrm{encoded}} = n_{\mathrm{keep}} + n_{\mathrm{reg}} = 85$ encoded tokens.

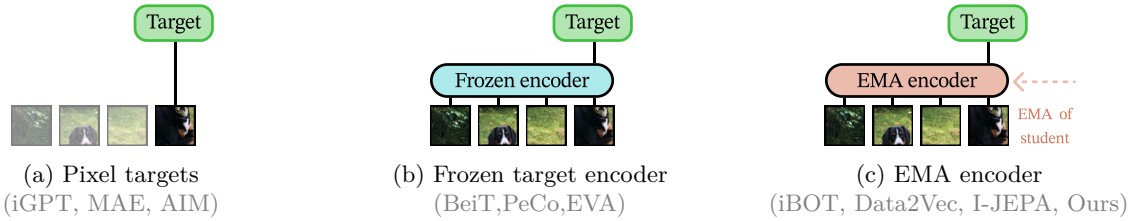

(a) Pixel targets
(iGPT, MAE, AIM)

(b) Frozen target encoder
(BeiT,PeCo,EVA)

(c) EMA encoder
(iBOT, Data2Vec, I-JEPA, Ours)

Figure 4: The target representations commonly used in MIM. (a) In the simplest case, the pixel values of the masked patches are flattened and directly used as targets. (b) An alternative is to use the output of a frozen encoder as targets, to obtain a more semantic representation. (c) To alleviate the need for a preexisting pretrained model, the target encoder can be an EMA of the online model. We focus on this case.

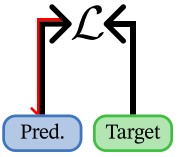
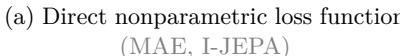

(a) Direct nonparametric loss function
(MAE, I-JEPA)

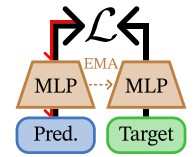

(b) Loss after MLP projection
(iBOT, DINOv2)

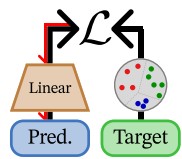

(c) Loss after clustering assignments
(proposed)

Figure 5: The different loss formulations considered here. We depict in red the flow of the gradient. (a) The loss can simply be a nonparametric function, such as MSE (in MAE) or Huber loss (in I-JEPA). (b) In iBOT and DINOv2, the prediction is passed through an MLP head, and the target is passed through an EMA of this head. The loss is the cross-entropy of their outputs. (c) We propose using a clustering to obtain the targets. The loss is the cross-entropy between the predictions and the assignments.

- We sample $n_{\mathrm{pred}} = 7$ coordinates among the dropped set, and for each we forward a `[MSK]` token through the predictor, which predicts their assignment by cross-attending to the encoded view.

A detailed visual diagram of the method can be found in fig. 2.

### 3.1 Clustering-based loss formulation

Some self-supervised learning approaches like SwAV and DINO can be interpreted as a form of latent clustering. They employ a cross-entropy loss between the student and teacher output distributions. These distributions, produced by a linear or MLP head, can be seen as soft cluster memberships, where the centroids correspond to the prototypes. Replicating the DINO objective, iBOT proposes to pass the student predictions through an MLP head and to pass the teacher embeddings through the EMA of this head. We argue that in iBOT, in the context of MIM, the clustering interpretation of the head is not valid any more, because the loss is not done between two versions of the same representation. While in DINO both targets and predictions are `[CLS]` tokens, in iBOT the targets are patch tokens while the predictions are special `[MSK]` tokens. This can also be seen as a *distribution mismatch*: the MLP head of the student is trained with `[MSK]` inputs but is instead applied to regular patch tokens in the teacher. This mismatch would be even stronger in an asymmetric architecture as in MAE or I-JEPA, where the targets and predictions come from two different networks (see the split design in fig. 6b). In practice, the iBOT loss is very unstable: without the stabilizing effect of the DINO loss, the iBOT formulation is unable to bootstrap itself and results in trivial representations (see table 1c). Our proposition is simple: to come back to a valid clustering interpretation and, hopefully, avoid such instabilities, we explicitly use a clustering on top of the EMA network to create the target representations. This decouples the training of the teacher projection from the student's and the training remains stable even in the absence of an additional global loss for stabilization.

**Online clustering.** Inspired by the minibatch $k$-means algorithm and the SwaV loss, we define our online clustering process as follows. Let $X \in \mathbb{R}^{n \times d}$ be the output of the teacher, with row vectors $x_i$ for $i \in \{1, \ldots, n\}$. We apply an L2 normalization and a linear projection to obtain the assignment logits $l_i \in \mathbb{R}^p$:

$$l_i = C \cdot \frac{x_i}{||x_i||}, \tag{1}$$

where $C$ is a matrix in $\mathbb{R}^{p \times d}$, whose rows are the $p$ "centroids" in dimension $d$. We then apply a softmax with temperature $\tau$ to obtain soft assignments:

$$a_i = \frac{\exp(l_i/\tau)}{\sum_{k=1}^{p} \exp(l_k/\tau)}. \tag{2}$$

We wish to estimate $C$ by solving the following problem:

$$\min_C \quad -\sum_{i=1}^{n} \sum_{k=1}^{p} a_i^{(k)} \log a_i^{(k)}, \tag{3}$$

where $a_i^{(k)}$ is the $k$-th coordinate in $a_i$. By minimizing the entropy, we force the assignments to be as close to one-hot as possible, which pushes the centroids towards their "assigned" samples. This can be seen as a form of clustering with a logistic loss. However, simply solving this problem can result in empty clusters. This is a common problem in mini-batch $k$-means (Sculley, 2010), and is usually solved by adding a reassignment phase: the centroids of the empty clusters are discarded and moved next to another non-empty centroid. In our case, we do not wish the centroids to move that abruptly, as it could disturb the training of the student. Instead, we propose to use the Sinkhorn-Knopp (SK) algorithm (Sinkhorn & Knopp, 1967), used in SeLa (Asano et al., 2020) and DINOv2 (Oquab et al., 2024). Using the SK algorithm, we obtain $a'$, an assignment where the distribution of tokens over the clusters is near-uniform:

$$\{a'_1, \ldots, a'_n\} \leftarrow \text{SK}(\{l_1, \ldots, l_n\}, \tau'), \tag{4}$$

with $\tau'$ another temperature parameter. Note that we do not backpropagate through the SK algorithm. We adapt the loss in eq. (3), and learn the centroids $C$ by minimizing the cross-entropy between $a$ and $a'$:

$$-\sum_{i=1}^{n} \sum_{k=1}^{p} a'^{(k)}_i \log a^{(k)}_i. \tag{5}$$

We minimize this loss with AdamW (Loshchilov, 2017) alongside the training of the main model. As the clustering loss and the MIM loss depend on two disjoint sets of parameters, we optimize them with two distinct optimizers without interference.

**Positional collapse.** A crucial point of self-supervised learning is avoiding trivial solutions, *i.e.* situations where the model minimizes the loss without learning useful features. These failure modes, also known as representation collapse, are usually addressed by adding regularization mechanisms. For example, color jittering (Chen et al., 2020b) helps prevent the model from focusing uniquely on color. While training CAPI, we observed a specific type of representation collapse as the positional encoding started outweighing the content of the patch embeddings. In the extreme case, the model learns to predict the position of the masked tokens instead of their content, resulting in a zero loss with a trivial model. In most observed cases, both content and position information are *entangled* in the target representation, while in the ideal scenario, the target would consist purely of semantic features. Although this problem does not arise in all training runs, it can be a significant issue when it does. We show in fig. 11 an example of feature maps of some early models affected by this phenomenon.

We propose a simple solution to alleviate the problem. By running the SK separately at each position in eq. (4), we force the joint distribution of tokens over positions and clusters to be near-uniform. Uniformity of the joint distribution directly implies zero mutual information between the clustering and the targets. This way, the modified SK ensures that our targets contain no positional information.

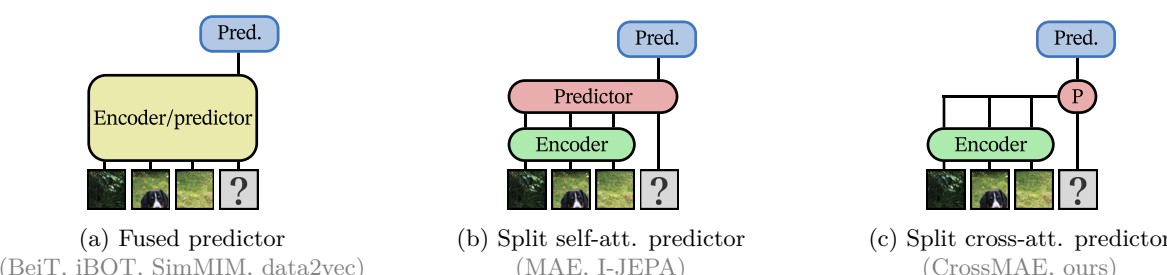

(a) Fused predictor
(BeiT, iBOT, SimMIM, data2vec)

(b) Split self-att. predictor
(MAE, I-JEPA)

(c) Split cross-att. predictor
(CrossMAE, ours)

Figure 6: The different predictor architectures discussed in the paper. Here, each box represents a transformer. The black lines represent the residual stream for a token. (a) In the fused architecture, the predictor and the encoder are the same transformer, which takes as input both the visible and mask tokens. (b) In the split architecture, the encoder processes only the visible tokens, while the predictor processes both the visible and the mask tokens. (c) The split architecture can be further refined by removing the visible tokens from the predictor. The predictor processes only the mask tokens, and uses cross-attention to access the context of the visible tokens.

## 3.2 Predictor architecture

Model architecture is another important component in reconstruction-based SSL. Two broad categories are widely used in previous work. BeiT, iBOT, and SimMIM use a *fused* architecture: a single vision transformer that takes as inputs patches and mask tokens (fig. 6a). Fused architectures are difficult to train and yield poor results, as reported in previous work (Zhou et al., 2022) and confirmed by our experiments. MAE uses a *split* architecture, with an encoder that only forwards the patches, saving memory and compute, and a predictor that forwards both patches and mask tokens (fig. 6b).

In this work, we use an even lighter predictor architecture, which was initially explored in crossMAE (Fu et al., 2024) for pixel-based reconstruction. In this case, the predictor forwards only the mask tokens (fig. 6c), which can access the context of the encoded patches via cross attention. Using a cross-attention predictor has two main advantages. First, it allows further efficiency gains, as we only forward a reduced set of tokens in the predictor, and we can even subsample this set of tokens. Second, mask tokens do not interact with each other in the cross-attention mechanism, *i.e.* each prediction is independent of other positions. This alleviates the need for multiple predictor forward passes with different prediction sets, as used in I-JEPA.

# 4 Experiments

In this section, we report empirical evaluations of our model. We describe experimental details and present some ablation studies. Then we discuss whole-image understanding results and dense prediction results.

## 4.1 Experimental setup

**Pretraining dataset.** Most methods from the self-supervised learning literature choose to pretrain on ImageNet-1k. This dataset is usually chosen because of its relatively small size, allowing for easy experimentation, and the ability to compare to existing methods pretrained on it. However, this has led to an overspecialization of SSL methods to the type of object-centric images found in ImageNet-1k. Recent foundation models obtain state-of-the-art results by exploiting much larger datasets, such as ImageNet-22k (Zhou et al., 2022) and LVD-142M (Oquab et al., 2024). If we are to design a method that can produce new foundation models, we believe it is crucial to design it to be able to handle such large datasets.

To this end, we carry out all ablation experiments on ImageNet-22k. It is composed of 14M images from 22k categories taken from the WordNet ontology. Although it is close to ImageNet-1k in nature, its much larger size and diversity make it suitable to train excellent foundation models, as reported by Oquab et al. (2024).

For our longer experiments, we train on multiple datasets: ImageNet-1k, for comparability with previous works, ImageNet-22k, to test scaling, Places205, to test training on more diverse and less object-centric data, and finally LVD-142M, a large-scale automatically curated dataset used in previous SSL foundation models. We refer the reader to Oquab et al. (2024) for more details on the curation process.

**Model architecture.** We do all our experiments with a Vision Transformer (Dosovitskiy et al., 2021) of 300M parameters (ViT-L). The model has a patch size of 14, a hidden size of 1024, and 24 transformer blocks. Note that in ablation experiments, the patch size used is 16. This architecture is widely used in various computer vision tasks, and most baselines provide a model of comparable size. We equip the vision transformer with registers (Darcet et al., 2024). These additional tokens were recently proposed as a way to add an information buffer, which enabled the model to produce smoother feature maps. For the predictor, we use 12 transformer blocks that cross-attend to the output of the encoder. This is similar to a standard transformer predictor (Vaswani et al., 2017), with the difference that we do not include self-attention layers. In this predictor, every token is forwarded independently and separately attends to the encoder output. When using a different encoder size, we align the embedding dimension, MLP ratio, and number of attention heads of the predictor to those of the encoder, and use a predictor depth equal to half that of the encoder.

**Implementation Details.** The learning rate follows a linear warmup followed by a cosine annealing. We truncate out the last 20% of the cosine, as proposed in I-JEPA (Assran et al., 2023). To simplify the choices

of parameters and schedules, we set the teacher EMA momentum to $\mu = 1 - lr$, and we set the learning rate for the clustering to half of the backbone learning rate. The impact of the most important hyperparameters will be discussed in section 4.2. All our pretraining hyperparameters are summarized in table 7.

**Evaluation protocol.** All the evaluations reported in this paper fall into two categories: image classification and semantic segmentation. For all classification tasks, we use an *attentive probe* (Assran et al., 2023; El-Nouby et al., 2024; Bardes et al., 2023). We use this evaluation protocol because our model does not learn a single global image representation, preventing the use of a linear probe. In this evaluation, we train an attentive pooling to extract a global vector and use this vector as input to a linear layer. The parameters of this probe are trained in a supervised fashion, and we report accuracy on the validation set. For segmentation tasks, we use lightweight classifiers on top of frozen local features. Previous works used a linear head trained with gradient descent on features of images augmented with various augmentations (Oquab et al., 2024). To obtain a more lightweight evaluation, we simply extract the features from all images in the dataset without augmentations, then train a linear logistic regression on these features with L-BFGS (Byrd et al., 1995) using an off-the-shelf library (Raschka et al., 2020). Although this results in lower mIoU numbers, the simplicity of the evaluation allows us to grid over different hyperparameters, producing very robust results. For an even more lightweight classifier, we also consider a non-parametric $k$-NN segmentation evaluation. For each test patch, we retrieve $k$ most similar patches in the training data and pool the segmentation label for that patch. We chose the optimal regularization parameters by doing a grid search using 10% of the training set. For all segmentation tasks, we measure performance using mIoU.

**Baselines.** We compare to the performance of previous models trained using missing context prediction: BeiT (Bao et al., 2021), MAE (He et al., 2022), data2vec 2.0 (Baevski et al., 2023), I-JEPA (Assran et al., 2023), and AIM (El-Nouby et al., 2024). Although AIM is not a masked image modeling method, we believe the similarity between masked image modeling and autoregressive modeling makes it a relevant comparison. To provide additional points of comparison, we report in grey the performance of iBOT (Zhou et al., 2022), MIM-refiner (Alkin et al., 2024) (dBOT-refined) and DINOv2+reg (Oquab et al., 2024; Darcet et al., 2024), who use a DINO or contrastive loss to complement a MIM objective. When multiple model sizes are available, we report the model closest in size to our models (ViT-L/14). In the specific case of MIM-refiner, multiple models were published with different MIM pretraining bases. We select the best-performing model on our evaluations, which turns out to be dBOT-refined in all cases. Finally, we include DINO, MoCov3 and MSN (Caron et al., 2021; Chen et al., 2021; Assran et al., 2022) as representative of non-MIM SSL methods.

## 4.2 Ablation Studies

We conduct extensive ablation studies to study the effect of design choices on performance. To make the ablation study more tractable, we train on the ImageNet-22k dataset for 100k iterations with a patch size of 16. To provide slightly more robust results, the default setting was run twice with different seeds, and the results of the two runs were averaged. All results are presented in table 1.

**Predictor architecture.** We evaluate the different predictor architectures. The split self-attention predictor produces better representations while training 32% faster than the fused predictor (table 1a). Using pure cross-attention in the predictor obtains even better representations and allows an additional 18% speedup by avoiding a forward on patch tokens. It also removes the dependency between different predictions, alleviating the need for repeated forwards of the predictor as in I-JEPA.

**Masking strategy.** The most common masking strategies in the literature are random masking, block masking (inpainting), or inverse block masking (outpainting). Inverse block masking induces a bias on the position of the masked patches: most often, the model will see the center of the image, and predict the edges. To prevent this, we propose applying a random circular shift to the mask before using it (`+roll`). This ensures that all positions in the image are equally likely to be masked. We ablate the type of masking in table 1b. The masking ratio is 65% for all strategies except random masking, where it is set to 90%, which increases performance. Random masking is much less effective than the other strategies, and inverse block masking works best, with a clear improvement when using `+roll`.

| | ADE | IN1k |
|---|---|---|
| Fused | 23.8 | 73.1 |
| Split, self-attn | 27.9 | 77.7 |
| Split, cross-attn | **29.1** | **81.4** |

(a) Predictor architecture

| | ADE | IN1k |
|---|---|---|
| random | 23.6 | 76.4 |
| block | 25.6 | 79.9 |
| inv. block | 27.2 | 80.7 |
| inv. block +roll | **29.1** | **81.4** |

(b) Masking strategy

| head | loss | ADE | IN1k |
|---|---|---|---|
| ∅ | I-JEPA | 23.7 | 79.3 |
| MLP | iBOT | 1.7 | 11.1 |
| MLP | CAPI | 26.4 | 80.8 |
| Linear | CAPI | **29.1** | **81.4** |

(c) Loss formulation

| | ADE | IN1k |
|---|---|---|
| $[0.2, 1.0]$ | 27.9 | 81.4 |
| $[0.6, 1.0]$ | **29.1** | **81.4** |
| $[1.0, 1.0]$ | 28.9 | 80.9 |

(d) Crop range

| | ADE | IN1k |
|---|---|---|
| 55% | 28.0 | 81.1 |
| 65% | **29.1** | **81.4** |
| 75% | 28.1 | 81.2 |

(e) Masking ratio

| depth | width | ADE | IN1k |
|---|---|---|---|
| 5 | 1536 | **30.9** | **81.5** |
| 12 | 1024 | 29.1 | 81.4 |
| 21 | 768 | 28.3 | 81.3 |

(f) Predictor shape

| | ADE | IN1k |
|---|---|---|
| 0 | 25.9 | 79.3 |
| 16 | **29.1** | **81.4** |

(g) Number of registers

| | ADE | IN1k |
|---|---|---|
| learnable | **30.0** | **81.6** |
| RoPE | 29.1 | 81.4 |

(h) Positional encoding

| | ADE | IN1k |
|---|---|---|
| Standard | 28.5 | 81.3 |
| Proposed | **29.1** | **81.4** |

(i) Sinkhorn-Knopp algorithm

Table 1: Ablation study of the main parameters and design choices in our algorithm. We report both image segmentation and classification. We highlight the default setting in gray, and bold the best-performing solution. We report ADE20K $k$-nn segmentation and ImageNet-1k attentive probing classification scores. An in-depth analysis of these results is provided in section 4.2.

**Loss formulation.** We evaluate the different strategies for computing a loss function discussed in section 3.1. We compare the performance of direct Huber loss, an iBOT loss with an MLP head, as well as our clustering-based loss with a linear or MLP head. With no head and direct loss, the model starts well but quickly regresses to worse representations. The iBOT head does not work in our setup, probably because of the split predictor design (since it implies that target tokens and prediction tokens do not come from the same network). Finally, the proposed clustering head alleviates all these issues and allows for stable training and good representations.

**Crop range.** To prevent overfitting, we use random cropping and flipping augmentations. We tweak the bounds of the cropping scale to $[c, 1.0]$ and try various $c$. We observe that the method can train well without any augmentation, resizing the training images to a fixed 224x224 resolution (80.9 on ImageNet with $c = 1.0$). We get better scores with minor cropping augmentation with a range of $[0.6, 1.0]$.

**Masking ratio.** In our model, we need to set the ratio $m$ of image patches that are masked, and reconstructed. We train our model for various $m$ and check the final performance. The optimal masking ratio seems to be around 0.65, but the algorithm seems quite robust to the choice of this parameter.

**Predictor shape.** We study the impact of the predictor depth on performance. We train the model with predictors of various depths and adapt the width to match the total number of parameters. Shallow networks will have a larger width. We see that a more shallow predictor leads to better performance, but in our informal experiments, we have observed that such architectures are less stable in long schedules. For this reason, we stick to the predictor with 12 layers and a width of 1024.

**Registers.** In our model, the local feature maps serve as a supervisory signal to train the model, so high-quality feature maps are crucial. To this end, we use register tokens, which were proposed to improve the quality of feature maps. We see that using registers has a large effect on performance, with an improvement of 3.2 points on ADE and 2.1 points on ImageNet when using 16 registers (table 1g). This is coherent with

| #prototypes | ADE | IN1k |
|---|---|---|
| 1024 | 14.9 | 73.8 |
| 2048 | 19.8 | 77.4 |
| 4096 | 27.5 | 80.7 |
| 8192 | 28.5 | 81.3 |
| 16384 | **29.1** | 81.4 |
| 32768 | **29.1** | **81.7** |

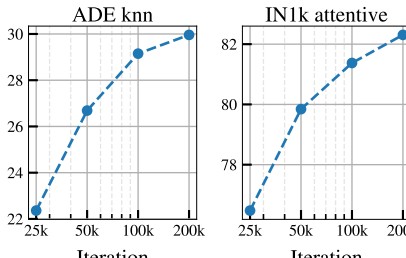

| dataset | ADE20K | IN1k |
|---|---|---|
| IN1k | 28.0 | **81.7** |
| IN22k | 29.1 | 81.4 |
| LVD-142M | **30.6** | 81.2 |

Figure 7: Additional ablation experiments. **(Left)** Influence of the number of prototypes. **(center)** Influence of the training length. Each point here is an independent training. **(right)** Influence of the training dataset.

recent observations on tabular data, for which registers were crucial to learn good representations. (Thimonier et al., 2024).

**Positional encoding.** In our experiments, we consider using different versions of positional encoding. We try the classic learnable position embeddings as well as relative ones such as RoPE (Su et al., 2024). Because of the ease of use and transferability to higher resolutions, we settle on using RoPE.

**Sinkhorn-Knopp algorithm.** We describe the problem of positional collapse in section 3.1. Changing the set of points considered in the Sinkhorn solves it, improving stability and granting a small performance increase (table 1i). We did not observe any positional collapse when using it.

**Number of prototypes.** We evaluate the effect of varying the number of prototypes $p$ in our clustering-based loss. The performance generally increases with the number of prototypes, at the cost of a higher memory footprint. We use $K = 16384$, which strikes a good balance between memory and performance.

**Scaling.** We study the effect of three scaling axes on the performance of our model: number of parameters, training length, and dataset size. To study the scaling potential of our algorithm, we train additional ViT-S and ViT-B models. We report the performance of the family of models in fig. 1. We see that performance consistently improves with model size, and for all models, our algorithm outperforms the state-of-the-art. In fig. 7 (center), we report the effect of the number of training iterations and pretraining dataset on performance. We train models using the ablation configs, on ImageNet-22k and with patch size 16. In fig. 7 (right), we show the influence of the training data on model performance. Using a larger dataset has a positive effect on segmentation, while only slightly deteriorating the ImageNet-1k accuracy.

## 4.3 Results

**Image classification.** We evaluate our model and compare it to state-of-the-art reconstruction-based SSL models. We run the evaluation on four datasets including object recognition, fine-grained classification, and scene recognition. We use ImageNet-1k (Russakovsky et al., 2015), iNaturalist 2021 (Van Horn et al., 2021), Places205 (Zhou et al., 2017), and SUN397 (Xiao et al., 2010). For ImageNet, we also report OOD robustness by running inference on additional test sets: ImageNet-V2 (Recht et al., 2019), ImageNet-ReaL (Beyer et al., 2020), ImageNet-A (Hendrycks et al., 2021), and ObjectNet (Barbu et al., 2019). For each model, we resize the image to 224×224 and collect the patch tokens output by the model. We feed those to an attentive probe implemented as a single layer of multi-head cross-attention with a single query (head size $d//64$, no residual). For $c$ classes and an embedding size $d$, the probe contains $2d^2 + (3 + c)d$ parameters, which are trained with AdamW for 10 epochs, selecting the best learning rate for each model/task on a held-out split of the training set. More details on the protocol are available in appendix H. All the results are summarized in table 2.

We see that our model outperforms all previous state-of-the-art models, by a large margin. When training on ImageNet-1k, we observe very good results, outperforming all other reconstruction-based models of

| Model | Arch. | Dataset | ImageNet | | | | | iNat | Places205 | SUN397 |
| | | | val | ReaL | v2 | A | ObjectNet | | | |
|---|---|---|---|---|---|---|---|---|---|---|
| DINO[†] | ViT-B/16 | IN1k | 76.9 | 83.6 | 65.3 | 14.3 | 34.8 | 64.1 | 59.0 | 61.2 |
| MoCov3[†] | ViT-B/16 | IN1k | 76.6 | 83.1 | 65.3 | 14.0 | 35.9 | 61.4 | 57.8 | 63.2 |
| MSN[‡] | ViT-L/16 | IN1k | 70.8 | 78.7 | 59.3 | 12.1 | 27.7 | 3.6 | 40.7 | 60.5 |
| iBOT | ViT-L/16 | IN1k | 80.9 | 86.5 | 70.3 | 41.9 | 28.9 | 70.5 | 62.0 | 64.6 |
| dBOT-refined | ViT-L/16 | IN1k | 82.8 | 87.9 | 73.4 | 44.7 | 49.0 | 72.9 | 63.2 | 68.7 |
| I-JEPA | ViT-H/14 | IN1k | 79.5 | 85.3 | 68.7 | 37.0 | 22.6 | 64.2 | 59.9 | 61.9 |
| MAE | ViT-L/16 | IN1k | 79.4 | 85.3 | 68.9 | 38.3 | 19.3 | 69.3 | 60.6 | 61.8 |
| Data2Vec 2.0 | ViT-L/16 | IN1k | 80.1 | 85.7 | 69.4 | 42.1 | 24.6 | 65.4 | 61.9 | 64.4 |
| CAPI | ViT-L/14 | IN1k | **82.9** | **87.6** | **72.9** | **47.5** | **43.7** | **76.8** | **65.4** | **70.9** |
| DINOv2+reg (distilled) | ViT-L/14 | LVD-142M | 86.5 | 89.9 | 79.1 | 75.5 | 64.5 | 86.6 | 68.3 | 79.6 |
| DINOv2+reg | ViT-g/14 | LVD-142M | 87.4 | 90.3 | 80.1 | 67.0 | 81.7 | 88.3 | 68.8 | 79.3 |
| BeiT | ViT-L/16 | IN22k | 40.8 | 46.1 | 30.7 | 8.7 | 2.1 | 26.5 | 36.8 | 29.9 |
| I-JEPA | ViT-H/14 | IN22k | 78.1 | 84.3 | 67.4 | 38.6 | 25.1 | 67.7 | 60.2 | 65.5 |
| AIM | ViT-600M/14 | DFN-2B+ | 79.0 | 84.8 | 67.9 | 41.0 | 20.4 | 73.5 | 62.5 | 66.1 |
| CAPI | ViT-L/14 | IN22k | 83.6 | 88.1 | 74.3 | 55.2 | 52.4 | **82.0** | 66.3 | 74.5 |
| CAPI | ViT-L/14 | Places205 | 79.2 | 84.7 | 68.4 | 39.1 | 33.0 | 73.4 | **68.6** | **77.5** |
| CAPI | ViT-L/14 | LVD-142M | **83.8** | **88.2** | **74.8** | **55.3** | **56.8** | 81.2 | 67.1 | 75.6 |

Table 2: Image classification results. We train an attentive probe on top of frozen features. For each baseline, we report the model closest in size to ViT-L/14. We separate models trained on ImageNet-1k only from models trained on a larger dataset. We report top-1 accuracy on all datasets, and for ImageNet we report some additional test sets. For reference, we report in grey the performance of other SSL models using DINO or contrastive losses. This shows that CAPI narrows the gap using only a MIM approach. †: Non-MIM SSL approaches. ‡: We report linear probing results (CLS token only) as they are better than attentive probing.

comparable size. CAPI excels particularly on out-of-distribution generalization, outperforming all baselines by more than 19 points on ObjectNet. Additionally, while the gap with other methods is somewhat limited on ImageNet, the difference in scene classification (SUN397) is much larger. Compared to iBOT, which uses a DINO loss to stabilize a MIM loss, CAPI performs better across the board. MIM-refiner, which uses a contrastive loss to refine a pretrained MIM model, obtains great scores when applied to dBOT. Our CAPI is comparable on most ImageNet benchmarks (+0.1 on IN-val, -0.3 on IN-ReaL, -0.5 on IN-v2) while being slightly better on ImageNet-A (+2.8) and falling behind on ObjectNet (-5.3). CAPI however significantly outperforms MIM-refiner out of domain, with a gap of 3.9, 2.2 and 2.2 points on iNaturalist, Places and SUN respectively.

Interestingly, we observe that CAPI works particularly well when trained on larger and more diverse datasets. In a second category, we compare models trained on datasets larger than ImageNet-1k. Our three CAPI models outperform all baselines on all datasets, except the CAPI-Places205 which is slightly below AIM on ImageNet-A. It should be noted, however, that AIM was trained on more than 2 billion images, with a sampling distribution tailored towards ImageNet. When trained on bigger datasets, CAPI significantly reduces the gap between reconstruction-based methods and our topline DINOv2+reg: the gap on ImageNet goes from 8.4 points to 3.6, and on SUN397 goes from 13.2 to 1.8.

**Dense image understanding.** As we have seen above, our model allows training high-quality local features that can be successfully pooled to solve image-level tasks. We also want to evaluate our model on dense prediction problems such as image segmentation. To this end, we run $k$-NN and linear segmentation following the protocol described in the experimental details. We run this for all the baselines reported above on three datasets. We use ADE-20k (Zhou et al., 2017), Pascal VOC 2012 (Everingham et al., 2010), and Cityscapes (Cordts et al., 2016). We report mIoU for all configurations in table 3.

As in the classification evals, CAPI trained on ImageNet-1k outperforms all reconstruction-based baselines by quite a wide margin on all evaluation setups. Compare to dBOT-refined, interestingly, CAPI obtains similar score in linear probing while having worse scores in $k$-nn probing. It is possible that the contrastive refining process is helpful to obtain a good $k$-nn score. When training on larger datasets, the conclusion is similar:

| Model | Arch. | Dataset | ADE-20k | | Pascal-VOC | | Cityscapes | |
|---|---|---|---|---|---|---|---|---|
| | | | $k$-NN | linear | $k$-NN | linear | $k$-NN | linear |
| DINO[†] | ViT-B/16 | IN1k | 20.1 | 25.2 | 48.8 | 56.1 | 32.0 | 35.4 |
| MoCov3[†] | ViT-B/16 | IN1k | 21.8 | 26.6 | 54.4 | 62.2 | 31.6 | 34.7 |
| MSN[†] | ViT-L/16 | IN1k | 18.3 | 22.7 | 47.4 | 55.8 | 28.8 | 34.2 |
| iBOT | ViT-L/16 | IN1k | 26.0 | 30.7 | 60.2 | 68.8 | 35.7 | 39.8 |
| dBOT-refined | ViT-L/16 | IN1k | 30.8 | 32.8 | 66.7 | 70.2 | 38.4 | 41.7 |
| I-JEPA | ViT-H/14 | IN1k | 20.8 | 25.7 | 56.7 | 63.6 | 26.4 | 34.5 |
| MAE | ViT-L/16 | IN1k | 21.5 | 27.4 | 53.7 | 61.5 | 32.8 | 38.5 |
| Data2Vec 2.0 | ViT-L/16 | IN1k | 24.2 | 27.6 | 57.5 | 58.1 | 32.8 | 38.2 |
| CAPI | ViT-L/14 | IN1k | **29.2** | **34.4** | **60.7** | **69.7** | **35.6** | **41.7** |
| DINOv2+reg (distilled) | ViT-L/14 | LVD-142M | 34.8 | 38.3 | 66.7 | 71.3 | 42.3 | 45.9 |
| DINOv2+reg | ViT-g/14 | LVD-142M | 34.0 | 39.0 | 63.0 | 72.8 | 42.0 | 46.8 |
| BeiT | ViT-L/16 | IN22k | 3.5 | 8.3 | 6.9 | 19.1 | 15.6 | 24.0 |
| I-JEPA | ViT-H/14 | IN22k | 18.9 | 26.3 | 55.0 | 64.2 | 23.2 | 34.2 |
| AIM | ViT-600M/14 | DFN-2B+ | 26.1 | 31.4 | 60.2 | 67.0 | 32.1 | 38.2 |
| CAPI | ViT-L/14 | IN22k | 29.7 | 35.2 | 61.1 | 70.4 | 35.2 | 41.0 |
| CAPI | ViT-L/14 | Places205 | **35.2** | **39.1** | 61.7 | 69.4 | **39.5** | **44.6** |
| CAPI | ViT-L/14 | LVD-142M | 32.1 | 37.2 | **63.8** | **72.7** | 38.9 | 44.3 |

Table 3: Semantic segmentation results. We train a $k$-NN or a logistic regression probe on top of frozen features. For each baseline, we report the model size closest to ViT-L/14. We separate models trained on ImageNet-1k only, and the ones trained on a larger dataset. We report mIoU on all datasets. For reference, we also report the performance of some other SSL models using DINO or contrastive losses to complement the MIM loss. This shows that CAPI narrows the gap using only a MIM approach. [†]: Non-MIM SSL approaches.

CAPI outperforms all baselines by a wide margin and even beats DINOv2+reg in some setups. When trained on Places205, CAPI achieves mIoU 1.2 points higher than DINOv2+reg on ADE20K $k$-nn segmentation. To our knowledge, this is the first time DINOv2+reg is bested on segmentation with frozen features on ADE20K. DINOv2+reg, however, remains the most versatile model, with good results across the board and the best results on Cityscapes.

## 4.4 Additional explorations

As a final set of experiments, we investigate some additional properties of our model. We investigate its robustness to change of input resolution and try to obtain global representations using the predictor.

**High-resolution image understanding.** Our model was trained on $224 \times 224$ images. To compare with the I-JEPA model trained natively at high resolution, we try to evaluate our model on $448 \times 448$ images. In table 4, we see that our model does not require high-resolution training or evaluation to achieve the best performance. Our model trained at 224 and evaluated at 224 outperforms the large I-JEPA model trained at 448 and evaluated at 448. Moreover, using RoPE embeddings, CAPI is more robust to resolution changes: it loses only 0.3% (versus 1.0% for I-JEPA) when increasing the resolution.

| Method | train res. | eval@224 | eval@448 |
|---|---|---|---|
| I-JEPA-H | 224 | 79.4 | 78.4 |
| I-JEPA-H | 448 | 79.6 | 82.5 |
| CAPI-L | 224 | **83.8** | **83.5** |

Table 4: ImageNet-1k attentive probing accuracy of I-JEPA and CAPI at different input resolutions.

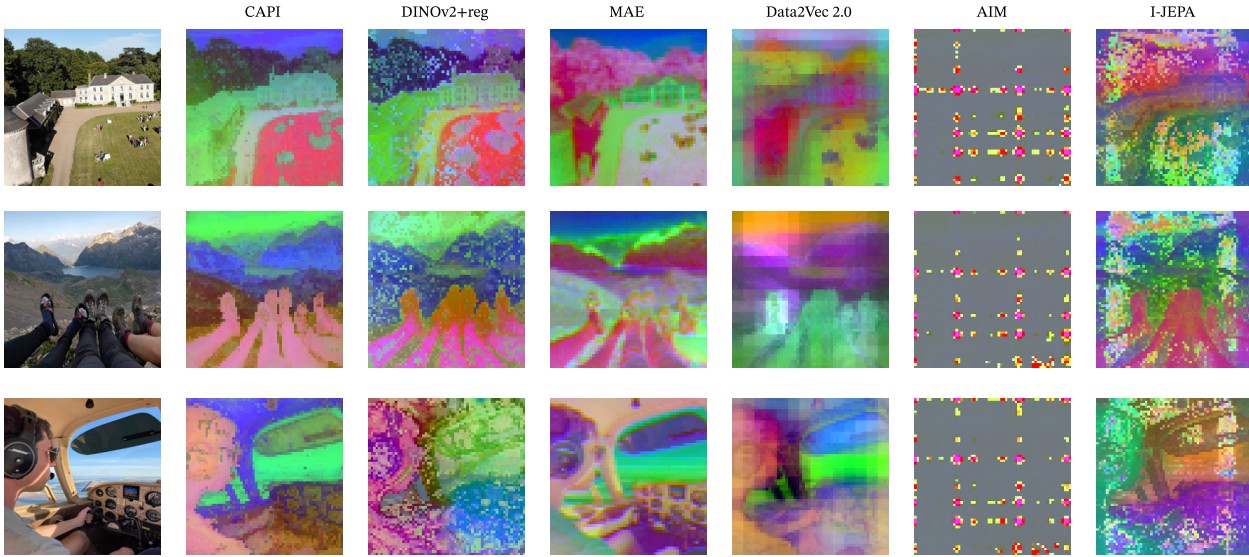

Figure 8: Visualization of the features of CAPI and baseline models. We apply a PCA to the features and map the three first components to RGB. The features produced by CAPI are discriminative and smooth.

|  | IN1k | | iNat21 | SUN397 |
| --- | --- | --- | --- | --- |
| Representation | $k$-NN | Linear | Linear | Linear |
| avg. pooling | 57.1 | 77.1 | 49.1 | 73.3 |
| predictor pooling | **73.8** | **81.1** | **69.6** | **77.4** |

Table 5: Classification using predictor representations, compared to the average pooling of the patch tokens.

**Qualitative feature analysis.** We propose a qualitative assessment of the dense features in fig. 8. The dense features computed with CAPI are amongst the most discriminative and smooth. We see the emergence of distinct objects, without much noise in uniform regions. We observe that the CAPI features are less noisy than the ones from DINOv2+reg, while being more focused on semantics and less on colors than the MAE features. For example, the shaded building in the first image has CAPI features similar to the other buildings, while in MAE the features are closer to other dark areas of the image.

**Obtaining global representations.** In all the previous evaluations reported in the experimental section, we trained a head on top of local features. Our model does not provide an aggregate representation like the `[CLS]` token in DINOv2. In this experiment, we try to exploit the predictor to obtain global image representations. We forward the whole image through the encoder, and then pass the same amount of mask tokens through the first attention layer of the predictor, cross-attending to the patch tokens. We obtain a global representation by average-pooling the output of this predictor attention layer. We train a linear model on this representation on several classification datasets and compare it with the average pooling of the patch embeddings in table 5. We see that using the attention pooling learned by the predictor provides better representations than averaging local features from the encoder.

# 5 Broader impact

As with any machine learning model, the impact of our work depends on how it is used. This work in particular focuses on improving the quality of self-supervised visual features, which can pose specific privacy issues if a model is trained on identifiable pictures, and pose fairness issues if the model absorbs biases from the dataset it is trained on. Additionally, the training of our models and the related exploratory experiments

directly use energy and computational resources, which can have an environmental impact. We discuss environmental impact in more detail in the appendix.

## 6 Discussion and Concluding Remarks

In this paper, we have proposed a novel reconstruction-based self-supervised learning algorithm. Our algorithm is based on an online clustering of dense features computed with a teacher network. The latent assignments are used as targets to train the student. We propose to implement the student as an encoder followed by a cross-attention predictor. The teacher is updated as an EMA of the encoder. The proposed algorithm is simple and allows the training of a state-of-the-art model in the category of reconstruction-based approaches. Our ViT-L outperforms all available reconstruction-based models, including much larger architectures. We have shown promising scaling trends until the 300M model sizes of the ViT family, opening up a potential for further scaling in future work. Some questions remain open: in particular, we believe it would be interesting to explore the scaling capabilities beyond the scale we explored, and to study the optimal training dataset for this sort of training. The question of whether such a pure reconstruction-based approach can reach the performance of contrastive learning methods remains open. We hope that our work will inspire further research in this direction.

### Acknowledgments

We thank Momo for fig. 8. Julien Mairal was supported by ERC grant number 101087696 (APHELEIA project).

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

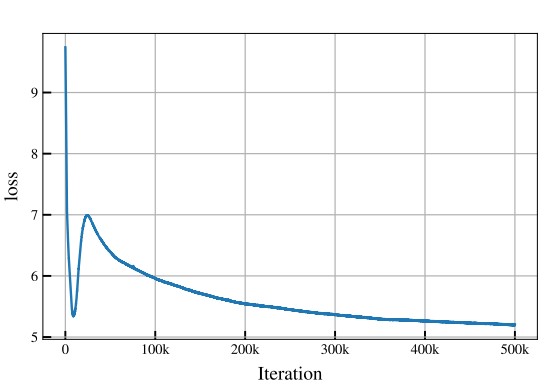

Figure 9: The loss curve of our CAPI ViT-L during training.

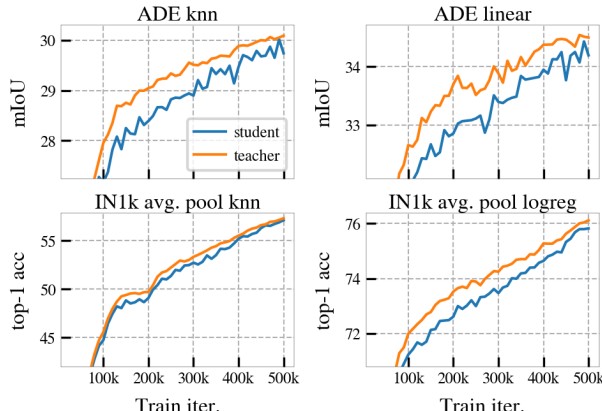

Figure 10: Comparative downstream scores of the teacher model and the student model throughout training.

## A  Loss curve

We report in fig. 9 the loss curve of our CAPI ViT-L model. After an initial adjustment period, the loss trends smoothly downwards, with no sign of instability or plateauing. Compared to other latent masked image modeling methods such as I-JEPA or iBOT, this trend is reassuring, and might indicate good potential for further scaling.

## B  Blockwise masking strategy

The so-called "block masking" strategy used in many masked image modeling methods is by no means standardizes and can actually refer to several different implementations. The most common block masking implementation was proposed in BeiT (Bao et al., 2021), then reused in iBOT (Zhou et al., 2022) and MAE (He et al., 2022). It involves sampling many rectangular regions and doing multiple attempts to mask out approximately the right number of patches. Another implementation was proposed in I-JEPA, adding multiple constraints on the masks, to obtain a similar multi-block mask. Additionally, some methods postprocess the proposed mask to obtain a constant number of masked patches, in order to keep the same sequence length in all batch elements.

In CAPI, we propose a simpler heuristic: we sample a single rectangular mask, and truncate out the excess patches at the lower right end. Conversely, our implementation of inverse block masking is to sample a block mask, then simply invert it.

## C  Batch size

In all our experiments, we use a batch size of 16384. Although the memory footprint is slightly offset by the high proportion of tokens not processed by the encoder and predictor, we recognize that this may be a limiting factor for some users. For the ViT-L, this batch size fits in 4 nodes of 8 A100 80GB GPUs. We believe that the batch size could be reduced to 4096 without significant loss in performance, but the recipe may need adjustments, and we do not have results at smaller batch size.

## D  Self-distillation interpretation

It was observed in DINO (Caron et al., 2021) that the downstream scores of the EMA model were consistently higher than the ones of the online model during training. This led to the interpretation of DINO as a

| Representation | Learnable head | IN1k | | | | | | |
|---|---|---|---|---|---|---|---|---|
| | | val | ReaL | V2 | A | R | Sketch | Objectnet |
| avg pooling | Linear | 76.8% | 82.5% | 64.4% | 21.8% | 43.5% | 33.9% | 33.7% |
| pred pooling | Linear | 81.8% | 86.9% | 71.6% | 46.7% | 55.8% | 47.0% | 48.3% |
| cat(pred pooling, avg pooling) | Linear | 81.9% | 87.0% | 71.5% | 45.8% | 55.6% | 46.5% | 47.1% |
| feature map | Attentive | 83.8% | 88.2% | 74.8% | 56.5% | 58.8% | 47.9% | 54.9% |

Table 6: Comparison of linear probing scores on ImageNet using different representations and classification heads. In the first three rows, the pooling is a fixed function, while in the fourth row, the pooling is learned specifically or ImageNet via an attention layer. We observe that, while the average pooling of CAPI does not produce a great global vector, the predictor pooling trick allows CAPI to be competitive on the ImageNet linear probing benchmark. The pooling, however, is not the optimal pooling for ImageNet since an attentive pooling manages to get a better score.

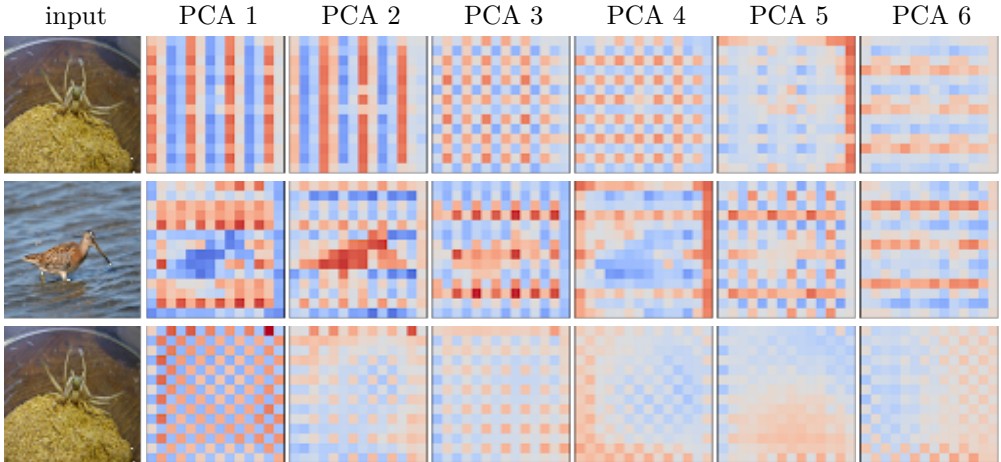

Figure 11: PCA visualization of the patch embeddings of some early CAPI models, showing partial positional collapse. The models did learn some useful nontrivial features (around 60% accuracy on ImageNet), but the positional information was very present and dominated the trainingn, preventing the model from learning better features.

self-distillation method, where the EMA model, the "teacher" distilled its slightly better representations into the online model, the "student". We observe that this interpretation still seems to hold in CAPI, albeit to a lesser extent, as evidenced by the comparison of teacher and student performance in fig. 10.

## E   Predictor pooling versus attentive probing

We report in table 6 the comparison of linear probing scores on ImageNet using different representations, with the topline being the attentive pooling results. We observe that the predictor learns a much better pooling than the baseline average pooling, although it does not reach the topline where the pooling is supervised (attentive probing).

## F   PCA visualizations of early models suffering positional from collapse

We show in fig. 11 the PCA visualization of the patch embeddings of an early CAPI model that showed some partial positional collapse.

```
3  M: Tensor  # b n d
4  M = M.exp()
5  for _ in range(3):
6-     M /= M.sum(dim=(0, 1))
7      M /= M.sum(dim=2)
```

(a) Standard SK

```
3  M: Tensor  # b n d
4  M = M.exp()
5  for _ in range(3):
6+     M /= M.sum(dim=0)
7      M /= M.sum(dim=2)
```

(b) Modified algorithm

Figure 12: PyTorch pseudo-code for the proposed modified Sinkhorn-Knopp algorithm. We normalize by the sum of the tokens for every given position, instead of normalizing across all positions.

| Hyperparameter | Value |
|---|---|
| Batch size | 16384 |
| Optimizer | AdamW |
| Learning rate | 1e-3 |
| Teacher momentum | $1 - lr$ |
| Clustering lr | $\frac{1}{2}lr$ |
| lr schedule | linear warmup + trunc. cosine |
| Warmup length | 10% |
| cosine truncation | 20% |
| Weight decay | 0.1 |
| AdamW $\beta$ | (0.9, 0.95) |
| Number of prototypes | 16384 |
| Student temperature | 0.12 |
| Teacher temperature | 0.06 |
| Num SK iter | 3 |
| Stochastic depth | 0.2 |
| Weight init | xavier_uniform |
| Norm layer | RMSnorm |
| Norm $\varepsilon$ | 1e-5 |
| Patch embed lr | $0.2 \cdot lr$ |
| Norm layer wd | $0.1 \cdot wd$ |
| Image size | 224 |
| Augmentations | RRCrop, HFlip |
| Training dtype | bf16 |
| Parallelism | FSDP |
| Pred. / im | 7 |
| Layerscale | No |
| Biases | No |
| Rope frequencies | logspace(7e-4, 7), axial |
| Masking type | inverse block+roll |
| Masking ratio | 65% |

Table 7: CAPI pretraining recipe

# G   Modified Sinkhorn-Knopp

We provide the pseudo-code for the standard Sinkhorn-Knopp and for our modified version in fig. 12. Both the original code and the proposed change are very simple. The actual code additionally contains an initial additive shift to prevent numerical instabilities in the exponential, as well as a collective `all_reduce` for distributed training.

# H    Detailed evaluation protocol

In all cases, our evaluations are performed with a frozen model, and use only the patch tokens outputted by the vision transformer. The input images are always at resolution 224×224.

## H.1    Classification

The backbone model is kept frozen, and we extract only the patch tokens from its output. On top of these features, we train an attentive pooling classifier, consisting of a learned query, two learned k anv v projections, and a final projection to the number of classes. The attention is multi-head, with the head dimension being fixed at 64 and the number of heads being $\frac{d_{model}}{64}$. This head is optimized with a cross-entropy loss and the AdamW optimizer (Kingma, 2014; Loshchilov, 2017) for 12500 iterations at batch size 1024 (10 ImageNet epochs). The learning rate is warmed up linearly for 1250 iterations then annealed with a cosine schedule. We grid the weight decay over (5e-4, 1e-3, 5e-2) and the peak learning rate over (1e-5, 2e-5, 5e-5, 1e-4, 2e-4, 5e-4, 1e-3, 2e-3, 5e-3, 1e-2), training one classifier head for each pair of hyperparameters (30 in total). We choose the optimal hyperparameters using the accuracy on a 10% held-out part of the training set, then finally report the accuracy of this classifier on the test set. The training dataset is lightly augmented using a `torchvision RandomResizedCrop` (maintainers & contributors, 2016) with default hyperparameters and a random horizontal flip. During evaluation, the images are resized to 256 then center cropped to 224×224 pixels.

## H.2    Segmentation

We compute the features for the train and test set considered at resolution 224, and hold out 10% of the train set as a validation set. Using these frozen features, the segmentation problem is reduced to a simple classification problem, on which we can use simple $k$-NN and linear classifiers. The linear classifier is trained for logistic regression with L-BFGS (Byrd et al., 1995) regularized with L2 penalty, as implemented in the `cuml` library (Raschka et al., 2020). In both cases, a grid of hyperparameters is tested, and the ones performing best on the validation set are retained. For the $k$-NN classifier, we grid the number of neighbors over (1, 3, 10, 30), and the distance used over (L2, cosine). For the linear classifier, we grid the regularisation parameter `C`, testing 8 values along a log-space between $10^{-6}$ and $10^5$.

## H.3    Feature standardization

Some of the baselines suffer from poor conditioning of their features, which can cause very bad results when fitting a logistic regression over these features. To reduce this issue, in the segmentation evaluation we standardize features by substracting their mean and dividing them by their standard deviation. These statistics are computed using the features from the training set only. This significantly improves the scores of pixel reconstruction-based methods, while the other methods are mostly unaffected. We report a comparison of segmentation results with and without standardization in table 8. In the rest of the paper, all segmentation results are obtained with standardization.

## H.4    Baselines

All baselines are vision transformers, allowing us to use the same evaluation protocol. We feed the 224×224 image to the model after imagenet normalization of the pixel values, and extract the patch tokens after the last transformer block. For the specific case of AIM (El-Nouby et al., 2024), we follow the advice from the original paper and extract the patch tokens after before the end of the ViT, specifically after layer 18.

# I    Compute cost and environmental footprint

We measure the training of a CAPI ViT-L model to take 180h on 32 A100 GPUs, amounting to 5763 A100 hours. This consumed around 2651 kWh of electricity, which we estimate to amount to approximately 928 kgCO2eq. The entire project used 3.75M A100 hours, which we similarly estimate to have emitted 604

| | | knn | | | logreg | | |
|---|---|---|---|---|---|---|---|
| model | standardization | ADE | Cityscapes | VOC2012 | ADE | Cityscapes | VOC2012 |
| CAPI | False | 32.5 | 39.2 | 64.9 | 37.9 | 44.7 | 73.2 |
| CAPI | True | 33.0 | 39.2 | 65.2 | 37.7 | 44.3 | 73.3 |
| aim 600M | False | 14.3 | 27.8 | 38.5 | 7.1 | 28.3 | 61.3 |
| aim 600M | True | nan | 32.1 | 60.2 | 31.5 | 38.2 | 67.0 |
| dinov2 vitg14+reg | False | 33.8 | 42.0 | 63.1 | 38.9 | 46.8 | 72.9 |
| dinov2 vitg14+reg | True | 34.0 | 42.0 | 63.0 | 39.0 | 46.8 | 72.8 |
| ijepa vith14 in1k | False | 20.1 | 25.9 | 57.4 | 25.2 | 33.5 | 63.0 |
| ijepa vith14 in1k | True | 20.8 | 26.4 | 56.7 | 25.7 | 34.5 | 63.6 |
| ijepa vith14 in22k | False | 17.9 | 22.9 | 56.2 | 24.1 | 32.8 | 63.6 |
| ijepa vith14 in22k | True | 18.9 | 23.2 | 55.0 | 26.4 | 34.2 | 64.2 |
| mae vitl16 | False | 7.8 | 25.3 | 17.3 | 27.4 | 38.4 | 61.5 |
| mae vitl16 | True | 21.5 | 32.8 | 53.7 | 27.4 | 38.5 | 61.5 |

Table 8: Comparison of segmentation results with and without standardization

Table 9: Summary of all models mentioned in the paper. We associate to each a unique uid, and detail the hyperparameters which are not constant across all runs.

| uid | dataset | #iter | patch size | enc depth | pred depth | enc dim | pred dim | lr | mom. | clust. lr | masking | ratio | roll | teacher head | student head | loss | pos. enc. | SK | #reg. | crop scale |
|---|---|---|---|---|---|---|---|---|---|---|---|---|---|---|---|---|---|---|---|---|
| Meb2b | LVD-142M | 500k | 14 | 24 | 12 | 1024 | 1024 | 1e-03 | 0.999 | 5e-04 | inv. block | 65% | True | clustering | Linear | CE | rope | modified | 16 | [60%,100%] |
| Mc2dd | LVD-142M | 500k | 14 | 12 | 6 | 768 | 768 | 1e-03 | 0.999 | 5e-04 | inv. block | 65% | True | clustering | Linear | CE | rope | modified | 16 | [60%,100%] |
| M8c4d | LVD-142M | 500k | 14 | 12 | 6 | 384 | 384 | 2e-03 | 0.998 | 1e-03 | inv. block | 65% | True | clustering | Linear | CE | rope | modified | 16 | [60%,100%] |
| Adcab | IN22k | 100k | 16 | 24 | 12 | 1024 | 1024 | 2e-03 | 0.996 | 1e-03 | inv. block | 65% | True | clustering | Linear | CE | rope | modified | 16 | [60%,100%] |
| Ae3f9 | IN22k | 100k | 16 | 24 | 12 | 1024 | 1024 | 2e-03 | 0.996 | 1e-03 | inv. block | 65% | True | clustering | Linear | CE | rope | modified | 16 | [60%,100%] |
| A0dd4 | IN22k | 100k | 16 | 24 | 12 | 1024 | 1024 | 2e-03 | 0.996 | 1e-03 | random | 90% | False | clustering | Linear | CE | rope | modified | 16 | [60%,100%] |
| A9b4a | IN22k | 100k | 16 | 24 | 12 | 1024 | 1024 | 2e-03 | 0.996 | 1e-03 | block | 65% | False | clustering | Linear | CE | rope | modified | 16 | [60%,100%] |
| A7cc0 | IN22k | 100k | 16 | 24 | 12 | 1024 | 1024 | 2e-03 | 0.996 | 1e-03 | inv. block | 65% | False | clustering | Linear | CE | rope | modified | 16 | [60%,100%] |
| A3bb3 | IN22k | 100k | 16 | 24 | 12 | 1024 | 1024 | 2e-03 | 0.996 | 1e-03 | inv. block | 65% | True | identity | identity | Huber | rope | standard | 16 | [60%,100%] |
| A2fcb | IN22k | 100k | 16 | 24 | 12 | 1024 | 1024 | 2e-03 | 0.996 | 1e-03 | inv. block | 65% | True | EMA | MLP | CE | rope | modified | 16 | [60%,100%] |
| Aeb48 | IN22k | 100k | 16 | 24 | 12 | 1024 | 1024 | 2e-03 | 0.996 | 1e-03 | inv. block | 65% | True | clustering | MLP | CE | rope | modified | 16 | [60%,100%] |
| A74f9 | IN22k | 100k | 16 | 24 | 12 | 1024 | 1024 | 2e-03 | 0.996 | 1e-03 | inv. block | 65% | True | clustering | Linear | CE | rope | modified | 16 | [100%,100%] |
| Ae7b3 | IN22k | 100k | 16 | 24 | 12 | 1024 | 1024 | 2e-03 | 0.996 | 1e-03 | inv. block | 65% | True | clustering | Linear | CE | rope | modified | 16 | [20%,100%] |
| A41b8 | IN22k | 100k | 16 | 24 | 12 | 1024 | 1024 | 2e-03 | 0.996 | 1e-03 | inv. block | 55% | True | clustering | Linear | CE | rope | modified | 16 | [60%,100%] |
| Ac8bc | IN22k | 100k | 16 | 24 | 12 | 1024 | 1024 | 2e-03 | 0.996 | 1e-03 | inv. block | 75% | True | clustering | Linear | CE | rope | modified | 16 | [60%,100%] |
| Af989 | IN22k | 100k | 16 | 24 | 5 | 1024 | 1536 | 2e-03 | 0.996 | 1e-03 | inv. block | 65% | True | clustering | Linear | CE | rope | modified | 16 | [60%,100%] |
| A2da0 | IN22k | 100k | 16 | 24 | 21 | 1024 | 768 | 2e-03 | 0.996 | 1e-03 | inv. block | 65% | True | clustering | Linear | CE | rope | modified | 16 | [60%,100%] |
| A9ce8 | IN22k | 100k | 16 | 24 | 12 | 1024 | 1024 | 2e-03 | 0.996 | 1e-03 | inv. block | 65% | True | clustering | Linear | CE | rope | modified | 0 | [60%,100%] |
| A1177 | IN22k | 100k | 16 | 24 | 12 | 1024 | 1024 | 2e-03 | 0.996 | 1e-03 | inv. block | 65% | True | clustering | Linear | CE | learn. | modified | 16 | [60%,100%] |
| A72fb | IN22k | 100k | 16 | 24 | 12 | 1024 | 1024 | 2e-03 | 0.996 | 1e-03 | inv. block | 65% | True | clustering | Linear | CE | rope | standard | 16 | [60%,100%] |
| M5e2e | IN22k | 500k | 14 | 24 | 12 | 1024 | 1024 | 5e-04 | 0.996 | 5e-04 | inv. block | 65% | True | clustering | Linear | CE | rope | modified | 16 | [60%,100%] |
| M2d34 | IN1k | 500k | 14 | 24 | 12 | 1024 | 1024 | 1e-03 | 0.999 | 5e-04 | inv. block | 65% | True | clustering | Linear | CE | rope | modified | 16 | [60%,100%] |
| M8319 | P205 | 500k | 14 | 24 | 12 | 1024 | 1024 | 1e-03 | 0.999 | 5e-04 | inv. block | 65% | True | clustering | Linear | CE | rope | modified | 16 | [60%,100%] |
| Abe05 | IN22k | 25k | 16 | 24 | 12 | 1024 | 1024 | 2e-03 | 0.996 | 1e-03 | inv. block | 65% | True | clustering | Linear | CE | rope | modified | 16 | [60%,100%] |
| Acd44 | IN22k | 50k | 16 | 24 | 12 | 1024 | 1024 | 2e-03 | 0.996 | 1e-03 | inv. block | 65% | True | clustering | Linear | CE | rope | modified | 16 | [60%,100%] |
| A2ca8 | IN22k | 200k | 16 | 24 | 12 | 1024 | 1024 | 2e-03 | 0.996 | 1e-03 | inv. block | 65% | True | clustering | Linear | CE | rope | modified | 16 | [60%,100%] |
| Aab94 | IN1k | 100k | 16 | 24 | 12 | 1024 | 1024 | 2e-03 | 0.996 | 1e-03 | inv. block | 65% | True | clustering | Linear | CE | rope | modified | 16 | [60%,100%] |
| Aa428 | LVD-142M | 100k | 16 | 24 | 12 | 1024 | 1024 | 2e-03 | 0.996 | 1e-03 | inv. block | 65% | True | clustering | Linear | CE | rope | modified | 16 | [60%,100%] |

tCO2eq for the electricity consumption. Note that the carbon footprint estimations here are purely scope 2 estimations, *i.e.* limited to electricity consumption, and are further limited to the electricity consumption of the GPUs. A full carbon accounting should additionally include many other harder to estimate emissions, such as the electricity consumption of the other server components and the rest of the datacenter appliances, and scope 3 emissions from the component manufacturing, datacenter construction, and their respective end-of-life. 2

## J List of models used

We provide in table 9 the list of all models presented in this paper, along with a unique hash identifier and the relevant hyperparameters. Non-listed hyperparameters are detailed in table 7. To disambiguate any possible unclarities in the presented results, table 10 provides the mapping from tables and figures to model identifiers.

Table 10: Reference of models used in different figures and tables.

| Fig | Models used |
|-----|-------------|
| fig. 1 | Meb2b, Mc2dd, M8c4d |
| table 1a | Adcab, Ae3f9, Aa5a3, A7d26 |
| table 1b | Adcab, Ae3f9, A0dd4, A9b4a, A7cc0 |
| table 1c | Adcab, Ae3f9, A3bb3, A2fcb, Aeb48 |
| table 1d | Adcab, Ae3f9, A74f9, Ae7b3 |
| table 1e | Adcab, Ae3f9, A41b8, Ac8bc |
| table 1f | Adcab, Ae3f9, Af989, A2da0 |
| table 1g | Adcab, Ae3f9, A9ce8 |
| table 1h | Adcab, Ae3f9, A1177 |
| table 1i | Adcab, Ae3f9, A72fb |
| table 2 | Meb2b, M5e2e, M2d34, M8319 |
| table 3 | Meb2b, M5e2e, M2d34, M8319 |
| table 4 | Meb2b |
| fig. 8 | Meb2b |
| table 5 | Meb2b |
| fig. 7 | Adcab, Ae3f9, Abe05, Acd44, A2ca8, Adcab, Ae3f9, Aab94, Aa428 |
| fig. 10 | Meb2b |
| fig. 14 | Meb2b |
| fig. 13 | Meb2b |

## K  Visualisations

In fig. 13 and fig. 14, we provide visualisations of the feature maps of CAPI compared to other state-of-the-art self-supervised vision models.

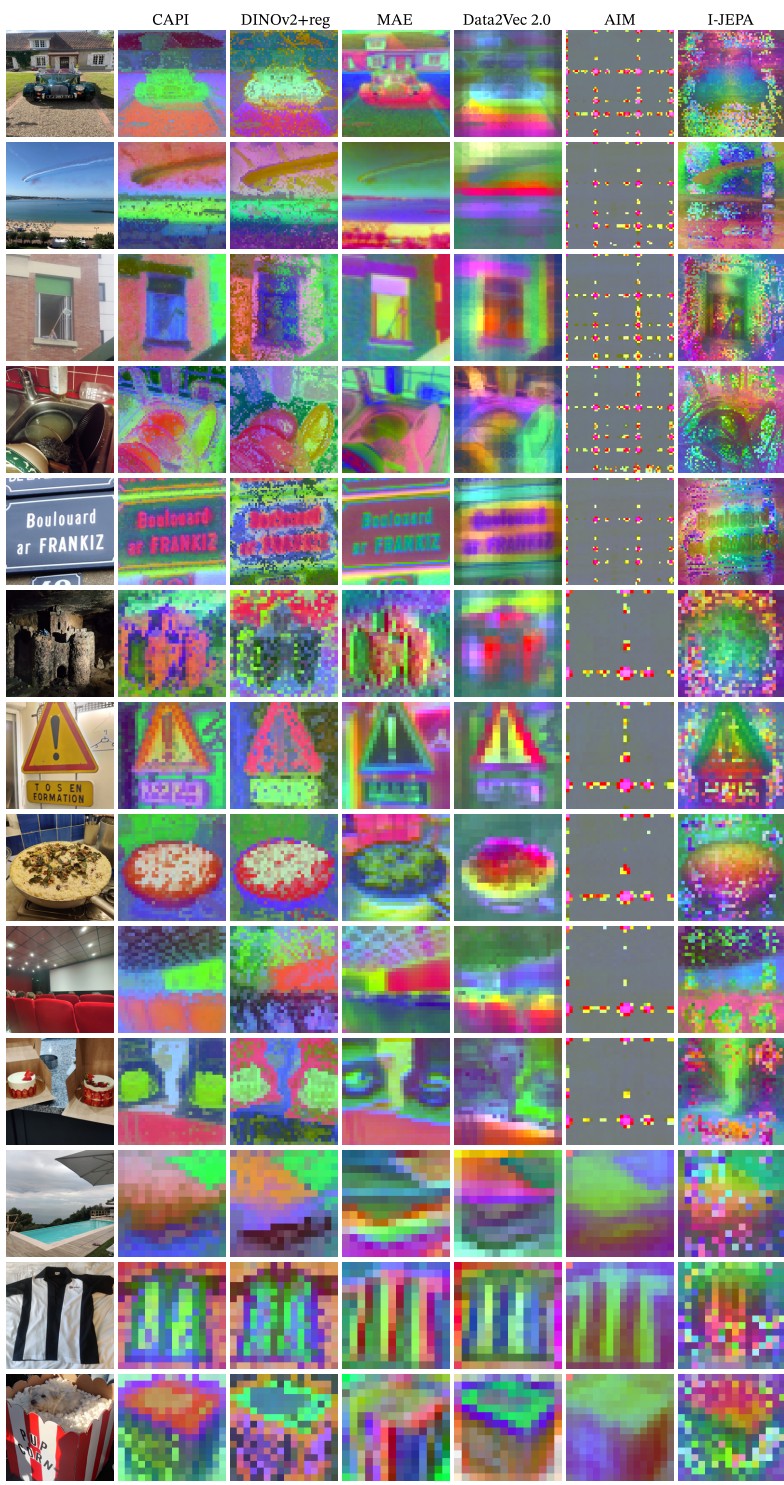

Figure 13: Visualization of the features produced by CAPI and other vision models at various resolutions: CAPI ViT-L/14, DINOv2+reg ViT-g/14 (Darcet et al., 2024), BEiT ViT-L/16 (Bao et al., 2021), AIM ViT-3B/14 (El-Nouby et al., 2024), MAE ViT-H/14 (El-Nouby et al., 2024), I-JEPA ViT-H/14 (Assran et al., 2023), and data2vec2 ViT-L/16 (Baevski et al., 2022). We apply a PCA decomposition to the dense outputs produced by each model for each image individually, and rescale the three first components to the RGB range for visualization.

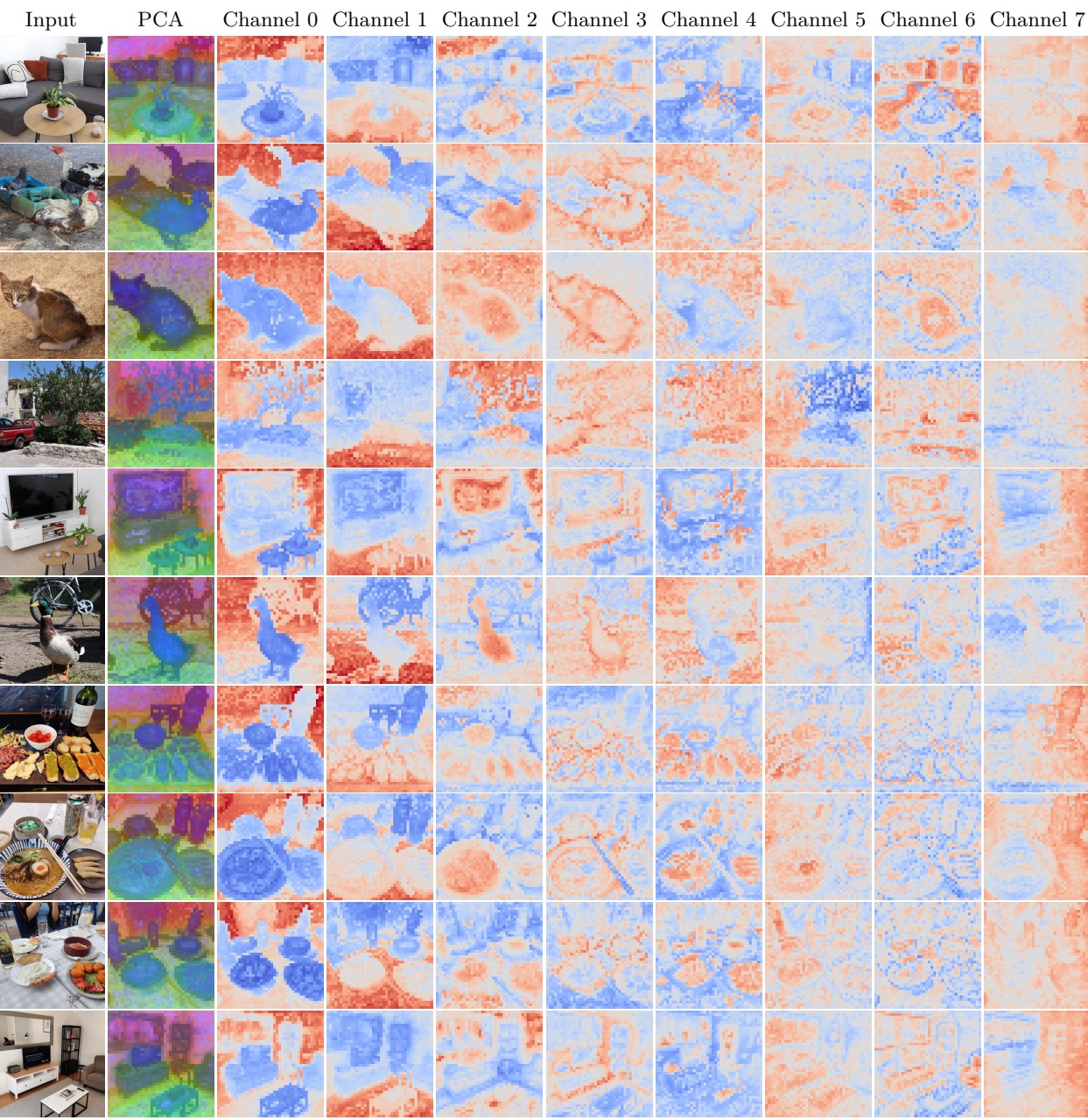

Figure 14: Visualization of the features produced by CAPI ViT-L/14 applied to images at 560 pixel resolution. We apply a PCA decomposition to the dense outputs produced by the model across all images. The first column shows the first 3 components as RGB. The next eight columns show the first eight channels individually using a `coolwarm` colormap from Matplotlib (Hunter, 2007).

