# OpenReview forum: "Cluster and Predict Latents Patches for Improved Masked Image Modeling"
_TMLR — Accepted by TMLR_

### Review · Reviewer_zw92 · 2025-03-06

**Summary Of Contributions:**

The authors outline an SSL learning framework for images which applies masked image modelling (MIM) in the latent space, positioning itself alongside I-JEPA as a contrastive approach for representations. Centrally, the authors are proposing a mini-batch clustering approach with modified Sinkhorn-Knopp (SK) iterations for doubly stochastic normalisation for the targets, with the main idea of applying latent clustering from epoch based assignments proposed in DeepCluster (Caron et al. 2018) to batch-based assignments via an EMA teacher. The work applies a CrossMAE decoder as a unidirectional prediction / representation head, which improves efficiency. The authors propose a custom patchwise / batchwise SK approach to prevent trivial correlations from positional encodings.

**Audience:**

Yes

**Broader Impact Concerns:**

This work advances the field of self-supervised modelling for images, and the ethical concerns related directly to the paper are arguably not specific enough to warrant a broader statement. The method does advance training of models on unlabelled data, but while there are potential concerns on privacy and intellectual property, there are existing methods that already enables self-supervised and unsupervised learning.

**Claims And Evidence:**

Yes

**Requested Changes:**

1. Adding baselines for MIM-Refiner and adding more discussion around latent versus autoencoding paradigms for MIM is necessary. This reviewer is aware that playing the "asking for more baselines"-card is straight out of the playbook of the "unreasonable reviewer". However in this case it would seem to be not only appropriate, but also serves to improve transparency and discourse in the field. I am not of the opinion that including these results (even if they are comparatively stronger) would lessen the contribution. On the contrary, it would strengthen the work.
2. The reliance on a massive batch size appears critical for stabilising the online clustering via SK iterations, but it could also be a limitation in resource-constrained settings. A more in-depth discussion how the models performance and stability might be affected if smaller batch sizes were used would be prudent. Adding a discussion whether there are strategies to mitigate this reliance would strengthen the work.
3. The patch dimension used in the model seems to be implied to be 16 x 16 (Overview of Training, Sec. 3), and 14 x 14 (ViT-L/14) in results (Tables 2, 3). For 224 x 224 images, a patch size of 14 would result in 256 patches, not 196. This leaves the reader unsure that the authors are consistent with previous works on their use of architectural notation, i.e. if the examples used in calculations in Sec. 3 are mistaken or hypothetical, to be taken outside the context of the modelling approach in the paper. This needs to be clarified.
4. The draft should define a clearer nomenclature for clustering on different levels, e.g. EMA clustering with DINO, direct epoch-based clustering in DeepCluster, and the batch-wise clustering approach presented in the paper.
5. The taxonomy outlined for different SSL approaches and clustering could be tightened, which would strengthen the clarity of the work.
6. Figure 8 comes across as much more informative than Figures 3, 4, by virtue of showing how the "split" decoder is applied as a variant of a representation head (e.g., SimCLR, DINO, etc.). This should perhaps be promoted to feature in the main paper, as opposed to a mention in the appendix. The roles of the two CE's are a little unclear, particularly the one used in the clustering process.

**Strengths And Weaknesses:**

**Strengths**:
1. The work introduces several mechanisms for improving efficiency for SSL in image data, achieving strong results. The focus on scalable SSL is a welcome objective that can potentially benefit the field as a whole, and could help democratise AI by enabling better training paradigms for researchers with limited compute budgets.
2. The work shows that latent MIM approaches yield competitive results. This seems to be a potentially fruitful area of novel research, and the paradigm on latent versus autoencoded masked image modelling serves as productive discourse with high interest and potential for advancing the field.
3. The authors use existing modelling techniques in a novel manner, and several of the presented ideas and proposals come across as intuitive and well aligned with explicit (and implicit) objectives. Individual contributions, such as custom SK normalisation to prevent positional collapse, cross-attention decoders as prediction / representation heads, and batch-wise clustered targets, serve as novel architectural designs in their own right.
4. There is a level of methodological rigour in the experimental results. Experimental ablations come across as meaningful and well constructed, and the results are presented clearly.

**Weaknesses**:
1. A central weakness in the current draft is that the work is somewhat opaque about the ideas, taxonomy, nomenclature, and contributions. The ambiguity in presentation and wording around how the work is placed in the greater taxonomy of SSL approaches for images weakens the clarity of the proposal, and makes the presented ideas less clear to the reader. More specifically;
    - (a). In the presentation, the exact nature of the “split” architecture is unclear. In CrossMAE, it seems that the decoder is designed via unidirectional cross-attention blocks to inform the predictions from latents without incurring the cost of predicting unmasked tokens, which are not aggregated as part of the loss function in autoencoding MIM approaches. As this reviewer understands the setup of the current work, the proposed “split” is applied as part of the encoder, since the predictions are made in the latent space directly without reconstruction. As such, the cross attention is now seemingly a part of an encoding process, distinct from how a decoder is viewed in the autoencoding setting (e.g., MAE). This could be made clearer by introducing Figure 8 earlier, or by clarifying the presentation of these ideas in context.
    - (b). Fig. 3 seems to be included to distinguish distillation from reconstruction, while Fig. 4 seems to look to distinguish training objectives, but without the distinction of latent / reconstruction and instead categorise approaches by design of trainable modules. The distinction comes across as somewhat arbitrary.
    - (c). It is implied that DINO is a latent clustering method (Sec. 3.1), but is not formulated as such in Fig. 4. This seemingly looks to setup the idea that clustering in the form of batch wise clustering “on-the-fly” is different from a "slow" EMA clustering approach used in DINO, but this distinction is not necessarily clear to the reader at a cursory glance. Again, this comes across as somewhat arbitrary.
    - (d). A minor nitpicky detail; presenting Sinkhorn-Knopp iterations for clustering as fundamentally based off SwAV (and DINOv2) is disputable, and it would seem that DeepCluster is a more fitting reference. Particularly as this work can -- at least in this reviewers opinion -- be seen as a continuation of DeepCluster.
2. The choice of baselines are missing certain key points of comparison. The choice of comparing with MAE and Data2Vec 2.0 when the current manuscript explicitly cites MIM-refiner comes across like a glaring omission. These results should be included and discussed to a greater extent, as they represent a compelling alternative in efficient SSL modelling approaches. Hence, positing the work in the broader discourse on representational MIM versus autoencoding MIM seems very much aligned with the objectives of the authors.
3. The method operates with a prohibitively large batch size (Table 6). While this improves stability in cluster assignments, it has a significant impact on efficiency and scalability, particularly in terms of memory.

---

> ### Author Response · Authors · 2025-03-21
>
> # Answer to reviewer zw92
>
>
> We thank reviewer zw92 for their comments.
> ## Clarification on the "split" architecture
> We understand now that our taxonomy has caused some confusion in the manuscript.
> The "split" architecture refers to using two separate vision transformers, one that processes the image patches (the encoder) and one that processes  the mask tokens (the decoder / predictor, depending on the nomenclature).
> This is in contrast with the "fused" architecture, where all tokens are processed by a single vision transformer, with no distinction between functional components
> To further clarify, what is called "encoder" in both CAPI and the baselines is only the first vision transformer, which will be used for downstream tasks.
> We refer to the second vision transformer in CAPI as the "predictor", to keep coherence with the I-JEPA nomenclature.
> Our predictor is the equivalent of the "decoder" in MAE and CrossMAE, in the sense that it takes as input encoded patch tokens output by the encoder, and makes a prediction for the loss and gets back gradients.
> After pretraining, however, in all cases only the encoder is kept for downstream tasks (except in our "predictor pooling" experiment).
> Additionally, we have sometimes used interchangeably the words "decoder" and "predictor" when talking about the CAPI predictor in the manuscript, which may have added some confusion.
> We revise the manuscript to only use "predictor" in the case of CAPI.
> Also following the reviewer's suggestion, we move figure 8 to the main part of the manuscript.
>
>
> ## Clarification on clustering, in particular in DINO
> We apologize for the confusion regarding DINO and clustering losses.
> We reformulated the caption of fig 4 (now fig 5) to make more explicit the differences between the three cases.
> The case 5.b. (middle) is about the fact that iBOT and DINOv2 project both prediction and target through an MLP before applying the loss function.
> We had previously called this case "DINO loss", but this was improper: we are not here talking about the DINO method, since it is not a MIM method. We changed this label to "loss after MLP projection".
> This projection head could, in the context of DINO, be considered an implicit clustering. We propose to instead explicitly use a clustering, clarifying the role of each component.
> ## Attribution of the use of SK
> To the best of our knowledge, DeepCluster does not employ the Sinkhorn-Knopp algorithm, as it uses a standard offline k-means and therefore does not need rebalancing.
> We believe our work is more heavily inspired by SwAV than DeepCluster, as the online clustering algorithm is very close to the one used in SwAV (without the swapping of assignments, since we are not learning a clustering invariant to augmentations).
> On the specific case of the use of SK for rebalancing in SSL, we observed that we had made a mistake though: it was introduced by SeLa, not by DeepCluster. We revised the manuscript to reflect that.
> ## baselines
> We thank the reviewer for the suggestion, MIM-refiner is definitely a valuable point of comparison, which we add to the manuscript. It provides strong points of comparisons for hybrid contrastive + MIM models trained on ImageNet-1k (82.8 IN1k attentive, and 30.8 ADE20K k-nn), complementing the iBOT model we were using as topline.
>
>
> ## batch size
> The batch size we used is indeed quite large, as this allowed us to speed up training.
> We believe the method could very probably work well with smaller batch sizes, as early experiments were performed at batch size 4096.However, the current hyperparameters are not tweaked to smaller batch sizes and therefore we cannot say how well the current recipe would perform at smaller batch sizes.
> Being unable to provide significant new experiments, we leave those explorations for future work.
> ## Requested changes
> - The MIM-refiner evaluations are quite lightweight, we added them to the tables
> - We added a discussion on batch size in the appendix, to clarify our choices.
> - Patch size: we apologize for the confusion regarding patch size. All ablations were conducted at patch size 16, while the training runs for the final models used patch size 14. We clarify that in the revised manuscript.
> - We reformulated the aspects around clustering in the manuscript, and hope they are now clearer.
> - We moved figure 8 to figure 2, in the main paper.
>
>
> We thank the reviewer again for pointing out parts of the manuscript that caused confusion or needed further clarification. This kind of feedback is indeed very valuable for authors.

---

### Review · Reviewer_SGLo · 2025-03-08

**Summary Of Contributions:**

The paper analyzes the limitations of existing masked image modeling (MIM) approaches from three perspectives -- target representation, loss formulation and predictor model architecture. Based on this analysis, it proposes a novel clustering-based MIM training framework, CAPI. CAPI outperforms previous MIM models on both image classification and segmentation tasks, significantly narrowing the performance gap with DINO v2, the current state-of-the-art self-supervised learning method. Extensive ablation studies validate the authors' design choices and provide several key insights for MIM training.

**Audience:**

Yes

**Broader Impact Concerns:**

I don't believe there are ethical concerns about this paper. Please discuss the limitation of the proposed method and add the Broader Impact Statement section in the final draft.

**Claims And Evidence:**

Yes

**Requested Changes:**

I have a few concerns regarding missing details and the clarity of some explanations in the paper. I would like to hear the authors' responses on these issues:
- Strictly speaking, AIM is not an MIM model. It is an autoregressive model using a prefix masking strategy. That's why AIM is able to scale up to 7B parameters. Could the authors clearly distinguish these two objectives or explain why AIM was categorized as an MIM model?
- The paper clearly demonstrates that CAPI outperforms previous MIM approaches. However, it seems less evident in what aspects CAPI outperforms DINO v2, if any. Are there specific advantages of CAPI over DINO v2 that the authors would like to emphasize? If relevant experimental results are available, please include them in the response.
- Please explicitly state the final total loss including both backbone training loss and clustering loss (Eq. 5). Are these two losses simply summed with equal weights?
- If possible, please elaborate further on the curation process of Loca.
- Could the authors compare the attentive probe (Table 1-2) and attention pooling with predictor (Table 5) under the same training configuration, such as the same model size and training set?
- In Table 1, are the evaluation results on ADE obtained using a KNN classifier, like the plot in Figure 6 (center)?
- In Table 1 (c), the authors observed that a linear head performs better than an MLP head under the CAPI loss formulation. Could the authors elaborate on possible reasons or insights behind this observation?
- How many training iterations were conducted for CAPI models reported in Table 3?

Minor typo: In Section 4.4, the performance drop when increasing the resolution is 0.3%, not 0.5%, for CAPI.

**Strengths And Weaknesses:**

Strengths
- The systematic analysis of masked image modeling with illustrative figures helps clarify the shortcomings of previous approaches and naturally highlights the advantages of CAPI.
- The paper provides extensive ablation studies to validate the design choices made for CAPI, presenting key findings and revealing critical factors for effective MIM training.
- The proposed CAPI outperforms previous MIM approaches on both high-level semantic (image classification) and dense prediction (segmentation) tasks.

Weaknesses: No major weaknesses warranting rejection. Please address the minor concerns below.

---

> ### Author Response · Authors · 2025-03-21
>
> Answer to reviewer SGLo
> We thank reviewer SGLo for their comments.
> ## AIM is not a MIM model
> The review points out, correctly, that AIM is not based on masked image modeling. We apologize for the confusion and we try to better motivate our choice of including AIM among the baselines..
> AIM is indeed an autoregressive model, and our study focuses on masked image modeling, i.e. predicting all missing patches at once and not autoregressively. Nonetheless, we believe it is a relevant baseline as belonging to a more broad class of models _trained to predict missing context_. In the revised manuscript, we added a sentence to clarify this point.
> ## Does CAPI outperform DINOv2
> In the general case, CAPI remains slightly weaker than DINOv2.
> However, there are instances in which CAPI does outperform DINOv2. For example, when trained on Places205, CAPI outperforms DINOv2+reg-giant on ADE20K segmentation (knn and linear), and when trained on Loca, CAPI outperforms DINOv2+reg-giant on Pascal VOC knn segmentation. It is possible that with further tuning and more extensive training and scaling, CAPI could surpass DINOv2.
> At present time, we do not have, however,additional results to present to compare to DINOv2.
> ## Relation between the clustering loss and the MIM loss
> The clustering loss and the MIM loss are independently computed and optimized. The two losses are never  summed during optimization, as they depend on two disjoint sets of parameters and they are managed by two distinct optimizers.
> In practice, we first compute the clustering loss and do the backward pass to get the clustering gradients, then we compute the second loss and run another backward pass to obtain MIM gradients.
> Two different optimizers then manage the two sets of parameters, each doing a step at each iteration. In the revised manuscript, we made sure that this procedure is clear. Furthermore, the code release should help clarify any further confusion.
> ## Loca
> We add to the manuscript additional clarification about the data collection and curation.
> ## comparison between attentive probe and predictor pooling
> We added an appendix table comparing average pooling, predictor pooling and learned attentive pooling. The predictor pooling does not reach the scores of the attentive probing, but it does significantly close the gap.
>
>
> | Representation                 | Learnable head | val   | ReaL  | V2    | A     | R     | Sketch | Objectnet |
> |--------------------------------|----------------|-------|-------|-------|-------|-------|--------|-----------|
> | avg pooling                    | Linear         | 76.8% | 82.5% | 64.4% | 21.8% | 43.5% |  33.9% |     33.7% |
> | pred pooling                   | Linear         | 81.8% | 86.9% | 71.6% | 46.7% | 55.8% |  47.0% |     48.3% |
> | cat(pred pooling, avg pooling) | Linear         | 81.9% | 87.0% | 71.5% | 45.8% | 55.6% |  46.5% |     47.1% |
> | feature map                    | Attentive      | 83.8% | 88.2% | 74.8% | 56.5% | 58.8% |  47.9% |     54.9% |
>
>
> ## classifier used in table 1
> We clarify in the revision that we indeed used the knn classifier
> ## linear vs MLP prediction head
> It is hard to say exactly why one head performs better or worse than the other. A few intuitions:
> - The MLP head follows the DINO MLP design, reused in many SSL projection heads. It includes a strong bottleneck of dimension 256 before the last layer, which could hinder performance
> - The role of the SSL heads has been hypothesized to be of applying a "guillotine regularization" [1], i.e. disconnecting the downstream representations from the representation used for the pretext (pretraining) loss. In our case, such a disconnect might not be needed, because the encoder representations and the loss are already separated by the predictor, which might already act as a strong head.
> ## number of iterations
> All ablations were conducted using 100k iterations, while the final runs used 500k iterations. We refer the reviewer to table 6 and 8 for more details on all hyperparameters used.
> ## Typo
> Corrected, thank you.
> ## limitations and broader impact
> We added a statement to the manuscript describing potential privacy, fairness and environmental issues that could arise from our work. We also detailed a few limitations and potential explorations in the conclusion.
>
>
> [1] Bordes, Florian, et al. "Guillotine regularization: Why removing layers is needed to improve generalization in self-supervised learning." arXiv

---

### Review · Reviewer_ogTS · 2025-03-11

**Summary Of Contributions:**

This paper introduces CAPI, a masked image modeling (MIM) framework that improves upon current methods in the k-NN and linear evaluation scheme. The framework predicts dense cluster assignments from masked patches using an EMA architecture. This work adds many add-ons to the typical MIM scheme, such as masking strategy and registers, to achieve promising results for MIM.

**Audience:**

Yes

**Broader Impact Concerns:**

--

**Claims And Evidence:**

Yes

**Requested Changes:**

The following suggested changes and experiments must be made (more details see above):
- The suggested experiments above to measure the effectiveness of the proposed approach inside the MIM literature.
- Sufficient acknowledgement of relevant works [1-3] beyond the related works section as the proposed method is an extension of [1] in a dense manner and [3] for images rather than videos.

These changes are critical for securing my recommendation. Given those, I believe that the paper is of interest to the communities based on the good results for MIM using k-NN and linear. I am looking forward to the authors' response.

**Strengths And Weaknesses:**

Strengths:

- The results of this MIM approach for k-NN and linear evaluation are promising.
- The paper is well written and has good motivation in the introduction.
- Proposed approach seems stable across hyperparameters like crop range, etc.
- Detailed hyperparameters are released to allow for the reproduction of the work.

Weaknesses:
1.  **Unclear effectiveness of method**: The paper directly compares to previous works, e.g. MAE, in a linear and kNN setup. However, this paper introduces many minor things, e.g., changing masking strategy, using registers, predictor shape, and predictor architecture. From the ablations (Tab 1. ) the biggest improvements one can observe are: predictor architecture + 6 %, masking strategy 5.5%, registers + 3.2%. These are minor details (including predictor pooling), which I acknowledge to be important, but make the comparison with previous works like MAE impossible. The paper frames to be about cluster and predict, so how impactful is the cluster and predict part actually w/o the add-ons? How would the model perform if it replaced the sinkhorn loss with a normal L2 loss? How would MAE perform with this masking strategy, registers, predictor architecture, etc.? Without these results, one can't tell the effectiveness of the proposed approach.

2. **Missing full finetuning results:** It is commonly known that MAE performance for non-full finetuning achieves limited performance. As this paper introduces an improved MIM approach, it also needs to compare with the previous MIMs in the full finetuning setting to see if it also improves there or at least does not degrade.

3. **Missing discussion to highly relevant works**: The paper does not cite nor acknowledge the contributions from previous works that
such as:
   -  [1] Masked Siamese Networks (MSN) use "cluster and predict" like this paper, but on a course (CLS token), not a dense level for MIM. This paper can be phrased as an extension, just in a dense manner.
    - [2] Sela: the first paper introducing predicting Sinkhorn assignments for SSL that has been later used in methods like DINO and SWAV. As the whole approach relies on this method, it should be referenced and acknowledged.
   - [3] SIGMA is very similar in idea (see title of [3]), predicting Sinkhorn-Knopp cluster assignments in the dense feature space for masked video modeling. As far as I see, [3] does not use a symmetric architecture like in this paper, which uses a symmetric one with EMA, thus being more elegant. *However, the overarching idea of cluster and predict in a dense manner for masked modeling has been implemented and proposed in [3]* for videos, which limits the novelty of this paper and needs to be correctly referenced and acknowledged.

   -> In summary, this work can be phrased as a combination of [1] and [3], predicting sinkhorn cluster assignments [2] for images [1] but in a dense manner like [3].

4. **Missing results on various backbones:** Results are shown solely on ViT-Large, however, a new approach must also be evaluated on various backbones to show effectiveness not just in one setting. I recommend including results on ViT-Base at least as it is common in MIM.

5. **Positional collapse exists?!** The paper introduces a fix to the problem of positional collapse (end of 3.1). However, when looking at the ablation study, does this positional collapse exist? Without the proposed solution for this problem, the performance only degrades by 0.1% and 0.6% on IN1k and ADE.

6. **Feature evaluation via unsupervised part segmentation**: Rather than visualizing the features and qualitatively calling them the most smooth and discriminative, one could opt for quantitative results through the evaluation of unsupervised part segmentation like in [4].

[1] Assran, M., et al. Masked Siamese Networks for Label-Efficient Learning. ECCV 2022.

[2] Asano, Y. M. et al. Self-labelling via simultaneous clustering and representation learning. ICLR 2020.

[3] Salehi, M., et al. SIGMA: Sinkhorn-Guided Masked Video Modeling. ECCV 2024.

[4] Ziegler, A., et al. Self-Supervised Learning of Object Parts for Semantic Segmentation. CVPR 2022.

---

> ### Author Response · Authors · 2025-03-21
>
> # Answer to reviewer ogTS
> We thank reviewer ogTS for the comments. Here are our answers.
> ## Effectiveness of the method
> The main concern expressed in the review centers around the effectiveness of the method.
> We firmly believe that the experimental evidence included in the paper, in particular the classification and segmentation results (table 2-3) and the different ablations (table 1), support the effectiveness of our method.
> We hope the reviewer will agree that our method as a whole is effective based on the results in table 2 and 3. Then, since our method is composed of several components, we have conducted an extensive ablation study to better understand the effects of the main choices.
> For the specific question on the importance of the clustering loss, we refer the reviewer to table 1 (c).
> In this ablation, we observe that replacing our clustering loss with a direct loss, specifically a Huber loss as done in I-JEPA, leads to a 5.4 point decrease in ADE20K k-nn segmentation, and 2.1 point decrease in ImageNet classification.
>
>
> ## Fine-tuning results
> We agree that the main strength of MAE is in full fine-tuning setups, and that MAE models are not meant to be used off-the-shelf for feature extraction.
> In our work, we choose to focus on representation learning, and as such our evaluation protocol uses lightweight probes on frozen features, without fine-tuning the whole model.
> Frozen evaluations like the ones we do have the benefit of assessing the model in a more direct way. On the contrary, full-fine tuning merely tests a model's potential as weight initialization.
> We strongly believe that developing methods for obtaining good frozen visual representations is an important research objective, leading to high-impact projects in scenarios where fine-tuning is not possible (see for example Tolan et al 2024 on canopy height estimation using frozen SSL features).
>
>
> Tolan, Jamie et al. "Very high resolution canopy height maps from RGB imagery using self-supervised vision transformer and convolutional decoder trained on aerial lidar." Remote Sensing of Environment (2024)
>
>
> ## Relevant work
> We thank the reviewer for pointing out several works that are relevant to our project. We apologize for missing those earlier and we have now included them in the manuscript
> In summary, here are our positioning relative to those works:
> - MSN, positioning itself as an extension of DINO, proposes an entropy loss to replace the centering of DINO, and using random patch dropping as an additional augmentation and efficiency trick (as in MAE). As we do not use the entropy regularizer, and our use of patch dropping is directly influenced by MAE, we disagree about our work being closely related to MSN.
> - SeLa is a relevant reference, and we apologize for missing it earlier. We revise the manuscript to correctly attribute the idea of using the Sinkhorn-Knopp algorithm for clustering to SeLa, fixing the previous oversight.
> - SIGMA is a very relevant reference, and we sincerely thank reviewer ogTS for correcting us on this. Though similar, the idea in SIGMA is different from our approach, based on the choice of target encoder. In SIGMA, the target encoder is either a frozen DINO model, or a separate patch-wise MLP trained independently from the main model. Instead, we focus on training a single encoder whose EMA is used as target encoder. On the other hand, the use of a Sinkhorn-Knopp to balance out the assignments for the MIM objective is indeed related to our approach. In the reviewed manuscript, we now assign proper credit to SIGMA in this aspect.
>
>
>
>
>
>
> ## Other backbones
> The review mentions the lack of reported results on smaller backbones. We highlight the results in  figure 1, where we compare the performance of CAPI models trained on Loca with a ViT-B/14 and a ViT-S/14 encoder.
> As we observe on the plot, the ViT-B/14 outperforms most other models, even significantly larger ones, on segmentation, while being comparable on classification.
> This confirms that CAPI is a valid method at multiple sizes.
>
>
> ## positional collapse
> Positional collapse did not appear consistently in our experiments, and the fact that we use RoPE helps alleviate it even in the absence of our modified SK.
> However, we can attest to having observed partial positional collapse in the past, and this can negatively affect training.
> To give a better empirical sense of the effect this can have, we add to the appendix a figure showing some PCA of feature maps of some early CAPI models that suffered from partial positional collapse.
>
>
> ## Unsupervised part segmentation evaluation
> We agree that this evaluation could bring a relevant additional signal to the paper.
> Unfortunately, we are not able to add such significant experiments to the paper at this point.

---

> > ### Comment · Reviewer_ogTS · 2025-04-10
> > **Response**
> >
> > Thanks for the response. I am following up on "Effectiveness of Method." Many open questions/concerns remain.
> > - Why are the results of Dinov2 ViT-L not included in Table 2? What happens if you train DINOv2 on ViT-Large on the same data? How does it compare? Why does Fig. 1 not have results for DINOv2 ViT-L only ViT-H?
> > - Though Table 2 compares MAE with CAPI, the comparison is unfair as CAPI uses, e.g., a different kind of masking and predictor.
> > - Why is there no comparison with DINOv1? (especially as you confirmed having results for Vit-Base). For example, Table 2 or Table 3 top. DINOv1 already gets 80.1 with linear, so that it will be even better with attentive probing for imagenet1k.
> > - Why is MSN [1] not included in the tables, especially for dense tasks (see Tab. 3)? From the results in MSN, it seems to be better than iBOT, thus being a strong competitor for Tab. 3.
> > - Given your previous response, full finetuning results are not needed as "We strongly believe that developing methods for obtaining good frozen visual representations is an important research objective, leading to high-impact projects in scenarios where fine-tuning is not possible"
> >  **However, this paper does not compare k-nn or linear ImageNet1k performance with previous works, which is standard in the literature, see Table 3 in MSN [1].**  Why?! I see that even the predictor average pooling achieves the same performance as MocoV3 with 81%, according to Table 3 MSN [1], and DINOv1 ViT-B getting 80.1% (even without predictor averaging, just simple encoder CLS/averaging). Also, DINOv2 has 85+%.
> > - The paper still lacks full finetuning results, common in the literature, like MSN or MAE.
> > - Side question: What happens if you remove the head from the predictor?
> >
> > My main concern remains: The paper introduces a new approach: "Cluster and predict". Given my above concerns, it seems the paper lacks results and comparisons, e.g., DINOv1, MOCOv3, MSN, DINOv2 ViT-L, etc., making it appear more convincing. As a reviewer, I can't tell if this "Cluster and predict" approach via Sinkhorn is the main driver and, as it has been told (see Title), or, what I believe so, the more technical add-ons, e.g., type of masking, predictor shape, etc. If the latter, it should be shown that such procedures benefit various approaches, not just overengineered for this particular one. My current intuition is that this approach is strong for dense tasks (given the local nature of the loss function), see Tab. 3, while still being on par or worse than global SSL approaches on global tasks (e.g., linear ImageNet1k same as Mocov3).
> >
> > I am open to changing my mind and think that if the paper is adjusted, it can benefit the community. Thus, I am looking forward to the response to my comments.

---

### Public Comment · ~Swaraj_Nanda1 · 2025-11-03
**Two comments on the mechanisms by which the prototypes are learned**

The CAPI work is a remarkable improvement over the vanilla iJEPA approach and is a fundamentally important step in pure patch-token representation learning. I present two points of observations for your consideration:

(A) Regularization on the prototypes:
I notice that a considerable amount of effort went into making sure that prototype favoritism is avoided, i.e., by balancing assignments using Sinkhorn-Knopp iterations. However, I find that the methodology does not include any collapse prevention mechanisms among the prototypes themselves. This is one behavior found in the vanilla dino clustering layer and has been elaborated in a prior study (Govindarajan, Hariprasath, et al. "On partial prototype collapse in the dino family of self-supervised methods." arXiv preprint arXiv:2410.14060 (2024)). Note, I am not involved in this work.

What would be important is if there could be additional constraints placed on the prototypes: (a) if prototype count is smaller than the dimension of the individual prototype vector, one could consider an orthogonality constraint, this can be useful in natural images where indeed the class categories are vast, (b) one can consider a simple variance-covariance loss as in VIC-Reg, although it might be quite the journey in figuring out what the loss-weights could be, (c) one could consider a KoLeo regularization of the prototypes (presented in your official code as a weight matrix).

I find that this is the missing piece, a regularization on the prototypes themselves, and would greatly add value to the method and add towards interpretability of the prototypes themselves.

(B) Total number of prototypes:
I find that the number of prototypes chosen in your work is shockingly low. This is because one is quite used to seeing very large values (65536 or 131072 prototypes in dinov2 for example) when pure cls-token approaches are considered. One may expect that in natural images, the total number of prototypes could be much higher than one usually encounters in cls-token based approaches, and this is because we are dealing with dense patches within the broad category of natural images. A values such as 16384 would perhaps suffice for my own field of histopathology, so I wonder why the authors chose such low values for their experiments with natural images.

---

> ### Author Response · Authors · 2025-11-04
> **Answer**
>
> Hi Swaraj, I'm very glad you appreciated the paper. As a general answer, I'm not able to work further on this topic, so I won't make updates in the short-term. I however warmly encourage to work on it if you have any intuitions on ways to improve it, improvements are always welcome.
>
> A few more specific thoughts:
>
> (A) It's a fair point, I have heard of this paper. We had observed prototype collapse during the DINOv2 project, and it's the main reason we switched to sinkhorn-knopp instead of EMA centering. After this switch, I did not observe prototype collapse in DINOv2. I haven't checked in CAPI, but since it also uses SK, my intuition is that there shouldn't be any.
>
> (B) The choice for this is mostly motivated by computational constraints: since the embeddings are directly projected from dimension 1k to 16k, the prototype matrix is around 16M parameters (in DINO/iBOT there is a bottleneck layer before, which makes the head lighter. Using a bottleneck did not seem to work as well in CAPI). Using more than 16k proves very expensive, and did not improve results in my experiments, so I stayed with 16k.
>
> But again: all of this is empirical, can change with the setup, and I'm always happy to be proven wrong. If you experiment with it, don't hesitate to send me results you want to discuss!

---

### Decision · Action_Editor_snth · 2025-05-09

**Recommendation:** Accept with minor revision

**Comment:**

This paper systematically investigates three core components of Masked Image Modeling (MIM)—self-supervision targets, loss functions, and predictor architectures—within the context of self-supervised learning. Building upon these analyses, the authors propose CAPI, a more effective MIM framework. All reviewers agree that the experimental results support the authors' claims, and the topic will likely appeal to TMLR’s audience interested in self-supervised learning. Therefore, AE recommends acceptance of this submission.

That said, one common concern raised by the reviewers is the lack of comparison with recent self-supervised learning (SSL) baselines. AE suggests addressing this concern in the revision to strengthen the paper’s contributions and better demonstrate the effectiveness of the proposed framework.

Revision Suggestions:
- **Include missing Figure 2.** The current manuscript refers to Figure 2, but it is not present in the submission.
- **Incorporate comparisons with key SSL baselines.** Specifically, please consider including methods such as DINOv1, MoCo v3, MSN, and DINOv2 (ViT-L). Although some of these methods do not rely on MIM, comprehensive comparisons across a diverse set of SSL approaches are essential in the field of self-supervised learning.

**Audience:**

Yes.

**Claims And Evidence:**

Yes.